# Online Learning with Transductive Regret

**Mehryar Mohri**
Courant Institute and Google Research
New York, NY
mohri@cims.nyu.edu

**Scott Yang**[*]
D. E. Shaw & Co.
New York, NY
yangs@cims.nyu.edu

## Abstract

We study online learning with the general notion of *transductive regret*, that is regret with modification rules applying to expert sequences (as opposed to single experts) that are representable by weighted finite-state transducers. We show how transductive regret generalizes existing notions of regret, including: (1) external regret; (2) internal regret; (3) swap regret; and (4) conditional swap regret. We present a general and efficient online learning algorithm for minimizing transductive regret. We further extend that to design efficient algorithms for the time-selection and sleeping expert settings. A by-product of our study is an algorithm for swap regret, which, under mild assumptions, is more efficient than existing ones, and a substantially more efficient algorithm for time selection swap regret.

## 1 Introduction

Online learning is a general framework for sequential prediction. Within that framework, a widely adopted setting is that of prediction with expert advice [Littlestone and Warmuth, 1994, Cesa-Bianchi and Lugosi, 2006], where the algorithm maintains a distribution over a set of experts. At each round, the loss assigned to each expert is revealed. The algorithm then incurs the expected value of these losses for its current distribution and next updates its distribution.

The standard benchmark for the algorithm in this scenario is the *external regret*, that is the difference between its cumulative loss and that of the best (static) expert in hindsight. However, while this benchmark is useful in a variety of contexts and has led to the design of numerous effective online learning algorithms, it may not constitute a useful criterion in common cases where no single fixed expert performs well over the full course of the algorithm's interaction with the environment. This had led to several extensions of the notion of external regret, along two main directions.

The first is an extension of the notion of regret so that the learner's algorithm is compared against a competitor class consisting of *dynamic* sequences of experts. Research in this direction started with the work of Herbster and Warmuth [1998] on tracking the best expert, who studied the scenario of learning against the best sequence of experts with at most $k$ switches. This work has been subsequently improved [Monteleoni and Jaakkola, 2003], generalized [Vovk, 1999, Cesa-Bianchi et al., 2012, Koolen and de Rooij, 2013], and modified [Hazan and Seshadhri, 2009, Adamskiy et al., 2012, Daniely et al., 2015]. More recently, an efficient algorithm with favorable regret guarantees has been given for the general case of a competitor class consisting of sequences of experts represented by a (weighted) finite automaton [Mohri and Yang, 2017, 2018]. This includes as special cases previous competitor classes considered in the literature.

The second direction is to consider competitor classes based on *modifications* of the learner's sequence of actions. This approach began with the notion of *internal regret* [Foster and Vohra, 1997, Hart and Mas-Colell, 2000], which considers how much better an algorithm could have performed if it had

---

[*]Work done at the Courant Institute of Mathematical Sciences.

switched all instances of playing one action with another, and was subsequently generalized to the notion of *swap regret* [Blum and Mansour, 2007], which considers all possible in-time modifications of a learner's action sequence. More recently, Mohri and Yang [2014] introduced the notion of *conditional swap regret*, which considers all possible modifications of a learner's action sequence that depend on some fixed bounded history. Odalric and Munos [2011] also studied regret against history-dependent modifications and presented computationally tractable algorithms (with suboptimal regret guarantees) when the comparator class can be organized into a small number of equivalence classes.

In this paper, we consider the second direction and study regret with respect to modification rules. We first present an efficient online algorithm for minimizing swap regret (Section 3). We then introduce the notion of *transductive regret* in Section 4, that is the regret of the learner's algorithm with respect to modification rules representable by a family of weighted finite-state transducers (WFSTs). This definition generalizes the existing notions of external, internal, swap, and conditional swap regret, and includes modification rules that apply to expert sequences, as opposed to single experts. Moreover, we present efficient algorithms for minimizing transductive regret. We further extend transductive regret to the *time-selection* setting (Section 5) and present efficient algorithms minimizing *time-selection transductive regret*. These algorithms significantly improve upon existing state-of-the-art algorithms in the special case of time-selection swap regret. Finally, in Section 6, we extend transductive regret to the *sleeping experts* setting and present new and efficient algorithms for minimizing *sleeping transductive regret*.

## 2 Preliminaries and notation

We consider the setting of prediction with expert advice with a set $\Sigma$ of $N$ experts. At each round $t \in [T]$, an online algorithm $\mathcal{A}$ selects a distribution $\mathsf{p}_t$ over $\Sigma$, the adversary reveals a loss vector $\mathbf{l}_t \in [0,1]^N$, where $l_t(x)$ is the loss of expert $x \in \Sigma$, and the algorithm incurs the expected loss $\mathsf{p}_t \cdot \mathbf{l}_t$.

Let $\Phi \subseteq \Sigma^\Sigma$ denote a set of *modification functions* mapping the expert set to itself. The objective of the algorithm is to minimize its $\Phi$-*regret*, $\mathrm{Reg}_T(\mathcal{A}, \Phi)$, defined as the difference between its cumulative expected loss and that of the best modification of the sequence in hindsight:

$$\mathrm{Reg}_T(\mathcal{A}, \Phi) = \max_{\varphi \in \Phi} \left\{ \sum_{t=1}^{T} \mathbb{E}_{x_t \sim \mathsf{p}_t}[l_t(x_t)] - \mathbb{E}_{x_t \sim \mathsf{p}_t}[l_t(\varphi(x_t))] \right\}. \tag{1}$$

This definition coincides with the standard notion of *external regret* [Cesa-Bianchi and Lugosi, 2006] when $\Phi$ is reduced to the family of constant functions: $\Phi_{\mathrm{ext}} = \{\varphi_a \colon \Sigma \to \Sigma \colon a \in \Sigma, \forall x \in \Sigma, \varphi_a(x) = a\}$, with the notion of *internal regret* [Foster and Vohra, 1997] when $\Phi$ is the family of functions that only switch two actions: $\Phi_{\mathrm{int}} = \{\varphi_{a,b} \colon \Sigma \to \Sigma \colon a, b \in \Sigma, \varphi_{a,b}(x) = 1_{x=a}b + 1_{x=b}a + x1_{x \neq a,b}\}$, and with the notion of *swap regret* [Blum and Mansour, 2007] when $\Phi$ consists of all possible functions mapping $\Sigma$ to itself: $\Phi_{\mathrm{swap}}$. In Section 4, we will introduce a more general notion of regret with modification rules applying to expert sequences, as opposed to single experts.

There are known algorithms achieving an external regret in $O(\sqrt{T \log N})$ with a per-iteration computational cost in $O(N)$ [Cesa-Bianchi and Lugosi, 2006], an internal regret in $O(\sqrt{T \log N})$ with a per-iteration computational cost in $O(N^3)$ [Stoltz and Lugosi, 2005], and a swap regret in $O(\sqrt{TN \log N})$ with a per-iteration computational cost in $O(N^3)$ [Blum and Mansour, 2007].

## 3 Efficient online algorithm for swap regret

In this section, we present an online algorithm, FASTSWAP, that achieves the same swap regret guarantee as the algorithm of Blum and Mansour [2007], $O(\sqrt{TN \log N})$, but admits the more favorable per-iteration complexity of $O(N^2 \log(T))$, under some mild assumptions.

Existing online algorithms for internal or swap regret minimization require, at each round, solving for a fixed-point of an $N \times N$-stochastic matrix [Foster and Vohra, 1997, Stoltz and Lugosi, 2005, Blum and Mansour, 2007]. For example, the algorithm of Blum and Mansour [2007] is based on a meta-algorithm $\mathcal{A}$ that makes use of $N$ external regret minimization sub-algorithms $\{\mathcal{A}_i\}_{i \in [N]}$ (see Figure 1). Sub-algorithm $\mathcal{A}_i$ is specialized in guaranteeing low regret against swapping expert $i$ with any other expert $j$. The meta-algorithm $\mathcal{A}$ maintains a distribution $\mathsf{p}_t$ over the experts and,

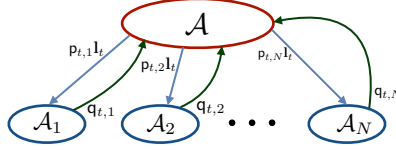

Figure 1: Illustration of the swap regret algorithm of Blum and Mansour [2007] or the FASTSWAP algorithm, which use a meta-algorithm to control a set of $N$ external regret minimizing algorithms.

---

**Algorithm 1:** FASTSWAP; $\{\mathcal{A}_i\}_{i=1}^N$ are external regret minimization algorithms.

---

**Algorithm:** FASTSWAP$((\mathcal{A}_i)_{i=1}^N)$

**for** $t \leftarrow 1$ **to** $T$ **do**
    **for** $i \leftarrow 1$ **to** $N$ **do**
        $\mathsf{q}_i \leftarrow \text{QUERY}(\mathcal{A}_i)$
    $\mathbf{Q}^t \leftarrow [\mathsf{q}_1 \cdots \mathsf{q}_N]^\top$
    **for** $j \leftarrow 1$ **to** $N$ **do**
        $c_j \leftarrow \min_{i=1}^N \mathbf{Q}_{i,j}^t$
    $\alpha_t \leftarrow \|\mathbf{c}\|_1; \quad \tau_t \leftarrow \left\lceil \frac{\log\left(\frac{1}{\sqrt{t}}\right)}{\log(1-\alpha_t)} \right\rceil$
    **if** $\tau_t < N$ **then**
        $\mathsf{p}_t \leftarrow \mathsf{p}_t^1 \leftarrow \frac{\mathbf{c}}{\alpha_t}$
        **for** $\tau \leftarrow 1$ **to** $\tau_t$ **do**
            $(\mathsf{p}_t^\tau)^\top \leftarrow (\mathsf{p}_t^\tau)^\top (\mathbf{Q}^t - \vec{1}\mathbf{c}^\top); \quad \mathsf{p}_t \leftarrow \mathsf{p}_t + \mathsf{p}_t^\tau$
        $\mathsf{p}_t \leftarrow \frac{\mathsf{p}_t}{\|\mathsf{p}_t\|_1}$
    **else**
        $\mathsf{p}_t^\top = \text{FIXED-POINT}(\mathbf{Q}^t)$
    $x_t \leftarrow \text{SAMPLE}(\mathsf{p}_t); \quad \mathbf{l}_t \leftarrow \text{RECEIVELOSS}()$
    **for** $i \leftarrow 1$ **to** $N$ **do**
        $\text{ATTRIBUTELOSS}(\mathsf{p}_t[i]\mathbf{l}_t, \mathcal{A}_i)$

---

at each round $t$, assigns to sub-algorithm $\mathcal{A}_i$ only a fraction of the loss, $(\mathsf{p}_{t,i}\mathbf{l}_t)$, and receives the distribution $\mathsf{q}_i$ (over the experts) returned by $\mathcal{A}_i$. At each round $t$, the distribution $\mathsf{p}_t$ is selected to be the fixed-point of the $N \times N$-stochastic matrix $\mathbf{Q}^t = [\mathsf{q}_1 \cdots \mathsf{q}_N]^\top$. Thus, $\mathsf{p}_t = \mathsf{p}_t\mathbf{Q}^t$ is the stationary distribution of the Markov process defined by $\mathbf{Q}^t$. This choice of the distribution is natural to ensure that the learner's sequence of actions is competitive against a family of modifications, since it is invariant under a mapping that relates to this family of modifications.

The computation of a fixed-point involves solving a linear system of equations, thus, the per-round complexity of these algorithms is in $O(N^3)$ using standard methods (or $O(N^{2.373})$, using the method of Coppersmith and Winograd). To improve upon this complexity in the setting of internal regret, Greenwald et al. [2008] estimate the fixed-point by applying, at each round, a single power iteration to some stochastic matrix. Their algorithm runs in $O(N^2)$ time per iteration, but at the price of a regret guarantee that is only in $O(\sqrt{N}T^{\frac{9}{10}})$.

Here, we describe an efficient algorithm for swap regret, FASTSWAP. Algorithm 1 gives its pseudocode. As with the algorithm of Blum and Mansour [2007], FASTSWAP is based on a meta-algorithm $\mathcal{A}$ making use of $N$ external regret minimization sub-algorithms $\{\mathcal{A}_i\}_{i \in [N]}$. However, unlike the algorithm of Blum and Mansour [2007], which explicitly computes the stationary distribution of $\mathbf{Q}^t$ at round $t$, or that of Greenwald et al. [2008], which applies a single power iteration at each round, our algorithm applies multiple *modified* power iterations at round $t$ ($\tau_t$ power iterations). Our modified power iterations are based on the REDUCEDPOWERMETHOD (RPM) algorithm introduced by Nesterov and Nemirovski [2015]. Unlike the algorithm of Greenwald et al. [2008], FASTSWAP uses a specific initial distribution at each round, applies the power method to a modification of the original stochastic matrix, and uses, as an approximation, an average of all the iterates at that round.

**Theorem 1.** *Let $\mathcal{A}_1, \ldots, \mathcal{A}_N$ be external regret minimizing algorithms admitting data-dependent regret bounds of the form $O(\sqrt{L_T(\mathcal{A}_i) \log N})$, where $L_T(\mathcal{A}_i)$ is the cumulative loss of $\mathcal{A}_i$ after $T$*

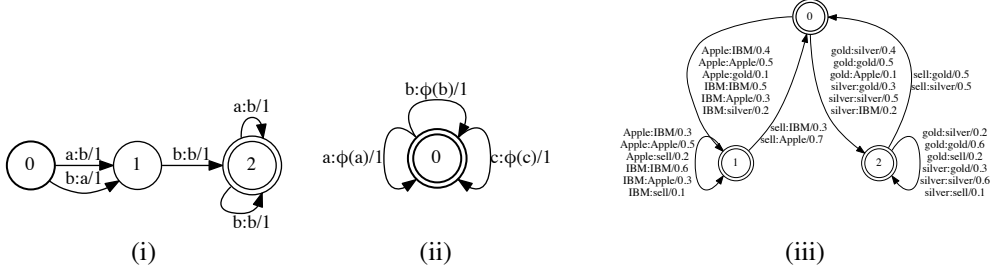

Figure 2: (i) Example of a WFST $\mathcal{T}$: $I_{\mathcal{T}} = 0$, $\mathsf{ilab}[\mathsf{E}_{\mathcal{T}}[0]] = \{a, b\}$, $\mathsf{olab}[\mathsf{E}_{\mathcal{T}}[1]] = \{b\}$, $\mathsf{E}_{\mathcal{T}}[2] = \{(0, a, b, 1, 1), (0, b, a, 1, 1)\}$. (ii) Family of swap WFSTs $\mathcal{T}_{\varphi}$, with $\varphi \colon \{a, b, c\} \to \{a, b, c\}$. (iii) A more general example of a WFST.

*rounds. Assume that, at each round, the sum of the minimal probabilities given to an expert by these algorithms is bounded below by some constant $\alpha > 0$. Then,* FASTSWAP *achieves a swap regret in* $O(\sqrt{TN \log N})$ *with a per-iteration complexity in* $O\big(N^2 \min\big\{\frac{\log T}{\log(1/(1-\alpha))}, N\big\}\big)$.

The proof is given in Appendix D. It is based on a stability analysis bounding the additive regret term due to using an approximation of the fixed point distribution, and the property that $\tau_t$ iterations of the reduced power method ensure a $\frac{1}{\sqrt{t}}$-approximation, where $t$ is the number of rounds. The favorable complexity of our algorithm requires an assumption on the sum of the minimal probabilities assigned to an expert by the algorithms at each round. This is a reasonable assumption which one would expect to hold in practice if all the external regret minimizing sub-algorithms are the same. This is because the true losses assigned to each column of the stochastic matrix are the same, and the rescaling based on the distribution $\mathsf{p}_t$ is uniform over each row. Furthermore, since the number of rounds sufficient for a good approximation can be efficiently estimated, our algorithm can determine when it is worthwhile to switch to standard fixed-point methods, that is when the condition $\tau_t > N$ holds. Thus, the time complexity of our algorithm is never worse than that of Blum and Mansour [2007].

## 4   Online algorithm for transductive regret

In this section, we consider a more general notion of regret than swap regret, where the family of modification functions applies to sequences instead of just to single experts. We will consider sequence-to-sequence mappings that can be represented by finite-state transducers. In fact, more generally, we will allow weights to be used for these mappings and will consider weighted finite-state transducers. This will lead us to define the notion of *transductive regret* where the cumulative loss of an algorithm's sequence of actions is compared to that of sequences images of its action sequence via a transducer mapping. As we shall see, this is an extremely flexible definition that admits as special cases standard notions of external, internal, and swap regret.

We will start with some preliminary definitions and concepts related to transducers.

### 4.1   Weighted finite-state transducer definitions

A weighted finite-state transducer (WFST) $\mathcal{T}$ is a finite automaton whose transitions are augmented with an output label and a real-valued weight, in addition to the familiar input label. Figure 2(i) shows a simple example. We will assume both input and output labels to be elements of the alphabet $\Sigma$, which denotes the set of experts. $\Sigma^*$ denotes the set of all strings over the alphabet $\Sigma$.

We denote by $\mathsf{E}_{\mathcal{T}}$ the set of transitions of $\mathcal{T}$ and, for any transition $e \in \mathsf{E}_{\mathcal{T}}$, we denote by $\mathsf{ilab}[e]$ its input label, by $\mathsf{olab}[e]$ its output label, and by $w[e]$ its weight. For any state $u$ of $\mathcal{T}$, we denote by $\mathsf{E}_{\mathcal{T}}[u]$ the set of transitions leaving $u$. We also extend the definition of $\mathsf{ilab}$ to sets and denote by $\mathsf{ilab}[\mathsf{E}_{\mathcal{T}}[u]]$ the set of input labels of the transitions $\mathsf{E}_{\mathcal{T}}[u]$.

We assume that $\mathcal{T}$ admits a single initial state, which we denote by $I_{\mathcal{T}}$. For any state $u$ and string $x \in \Sigma^*$, we also denote by $\delta_{\mathcal{T}}(u, x)$ the set of states reached from $u$ by reading string $x$ as input. In particular, we will denote by $\delta_{\mathcal{T}}(I_{\mathcal{T}}, x)$ the set of states reached from the initial state by reading string $x$ as input.

The input (or output) label of a path is obtained by concatenating the input (output) transition labels along that path. The weight of a path is obtained by multiplying is transition weights. A path from the initial state to a final state is called an *accepting path.* A WFST maps the input label of each accepting path to its output label, with that path weight probability.

The WFSTs we consider may be non-deterministic, that is they may admit states with multiple outgoing transitions sharing the same input label. However, we will assume that, at any state, outgoing transitions sharing the same input label admit the same destination state. We will further require that, at any state, the set of output labels of the outgoing transitions be contained in the set of input labels of the same transitions. This requirement is natural for our definition of regret: our learner will use input label experts and will compete against sequences of output label experts. Thus, the algorithm should have the option of selecting an expert sequence it must compete against.

Finally, we will assume that our WFSTs are stochastic, that is, for any state $u$ and input label $a \in \Sigma$, we have $\sum_{e \in \mathsf{E}_{\mathcal{T}}[u,a]} w[e] = 1$. The class of WFSTs thereby defined is broad and, as we shall see, includes the families defining external, internal and swap regret.

## 4.2 Transductive regret

Given any WFST $\mathcal{T}$, let $\mathcal{T}$ be a family of WFSTs with the same alphabet $\Sigma$, the same set of states $Q$, the same initial state $I$ and final states $F$, but with different output labels and weights. Thus, we can write $I_{\mathcal{T}}, F_{\mathcal{T}}, Q_{\mathcal{T}}$, and $\delta_{\mathcal{T}}$, without any ambiguity. We will also use the notation $\mathsf{E}_{\mathcal{T}}$ when we refer to the transitions of a transducer within the family $\mathcal{T}$ in a way that does not depend on the output labels or weights. We define the learner's transductive regret with respect to $\mathcal{T}$ as follows:

$$
\operatorname{Reg}_T(\mathcal{A}, \mathcal{T}) = \max_{\mathcal{T} \in \mathcal{T}} \left\{ \sum_{t=1}^{T} \mathop{\mathbb{E}}_{x_t \sim \mathsf{p}_t} [l_t(x_t)] - \sum_{t=1}^{T} \mathop{\mathbb{E}}_{x_t \sim \mathsf{p}_t} \left[ \sum_{e \in \mathsf{E}_{\mathcal{T}}[\delta_{\mathcal{T}}(I_{\mathcal{T}}, x_{1:t-1}), x_t]} w[e] \, l_t(\mathsf{olab}[e]) \right] \right\}. \quad (2)
$$

This measures the maximum difference of the expected loss of the sequence $x_1^T$ played by $\mathcal{A}$ and the expected loss of a competitor sequence, that is a sequence image by $\mathcal{T} \in \mathcal{T}$ of $x_1^T$, where the expectation for competing sequences is both over $\mathsf{p}_t$s and the transitions weights $w[e]$ of $\mathcal{T}$. We also assume that the family $\mathcal{T}$ does not admit proper non-empty invariant subsets of labels out of any state, i.e. for any state $u$, there exists no proper subset $\mathsf{E} \subsetneq \mathsf{E}_{\mathcal{T}}[u]$ where the inclusion $\mathsf{olab}[\mathsf{E}] \subseteq \mathsf{ilab}[\mathsf{E}]$ holds for all $\mathcal{T} \in \mathcal{T}$. This is not a strict requirement but will allow us to avoid cases of degenerate competitor classes.

As an example, consider the family of WFSTs $\mathcal{T}_a$, $a \in \Sigma$, with a single state $Q = I = F = \{0\}$ and with $\mathcal{T}_a$ defined by self-loop transitions with all input labels $b \in \Sigma$ with the same output label $a$, and with uniform weights. Thus, $\mathcal{T}_a$ maps all labels to $a$. Then, the notion of transductive regret with $\mathcal{T} = \{\mathcal{T}_a \colon a \in \Sigma\}$ coincides with that of external regret.

Similarly, consider the family of WFSTs $\mathcal{T}_{\varphi}$, $\varphi \colon \Sigma \to \Sigma$, with a single state $Q = I = F = \{0\}$ and with $\mathcal{T}_{\varphi}$ defined by self-loop transitions with input label $a \in \Sigma$ and output $\varphi(a)$, all weights uniform. Thus, $\mathcal{T}_{\varphi}$ maps a symbol $a$ to $\varphi(a)$. Then, the notion of transductive regret with $\mathcal{T} = \{\mathcal{T}_{\varphi} \colon \varphi \in \Sigma^{\Sigma}\}$ coincides with that of swap regret (see Figure 2 (ii)). The more general notion of *k-gram conditional swap regret* presented in Mohri and Yang [2014] can also be modeled as transductive regret with respect to a family of WFSTs ($k$-gram WFSTs). We present additional figures illustrating all of these examples in Appendix A.

In general, it may be desirable to design WFSTs intended for a specific task, so that an algorithm is robust against some sequence modifications more than others. In fact, such WFSTs may have been learned from past data. The definition of transductive regret is flexible and can accommodate such settings both because a transducer can conveniently help model mappings and because the transition weights help distinguish alternatives. For instance, consider a scenario where each action naturally admits a different swapping subset, which may be only a small subset of all actions. As an example, an investor may only be expected to pick the best strategy from within a similar class of strategies. For example, instead of buying IBM, the investor could have bought Apple or Microsoft, and instead of buying gold, he could have bought silver or bronze. One can also imagine a setting where along the sequences, some new alternatives are possible while others are excluded. Moreover, one may wish to assign different weights to some sequence modifications or penalize the investor for choosing strategies that are negatively correlated to recent choices. The algorithms in this work are flexible

enough to accommodate these environments, which can be straightforwardly modeled by a WFST. We give a simple example in Figure 2(iii) and give another illustration in Figure 5 in Appendix A, which can be easily generalized. Notice that, as we shall see later, in the case where the maximum out-degree of any state in the WFST (size of the swapping subset) is bounded by a mild constant independent of the number of actions, our transductive regret bounds can be very favorable.

### 4.3  Algorithm

We now present an algorithm, FASTTRANSDUCE, seeking to minimize the transductive regret given a family $\mathcal{T}$ of WFSTs.

Our algorithm is an extension of FASTSWAP. As in that algorithm, a meta-algorithm is used that assigns partial losses to external regret minimization slave algorithms and combines the distributions it receives from these algorithms via multiple reduced power method iterations. The meta-algorithm tracks the state reached in the WFST and maintains a set of external regret minimizing algorithms that help the learner perform well at every state. Thus, here, we need one external regret minimization algorithm $\mathcal{A}_{u,i}$, for each state $u$ reached at time $t$ after reading sequence $x_{1:t-1}$ and each $i \in \Sigma$ labeling an outgoing transition at $u$. The pseudocode of this algorithm is provided in Appendix B.

Let $|\mathsf{E}_{\mathcal{T}}|_{\mathsf{in}}$ denote the sum of the number of transitions with distinct input label at each state of $\mathcal{T}$, that is $|\mathsf{E}_{\mathcal{T}}|_{\mathsf{in}} = \sum_{u \in Q_{\mathcal{T}}} |\mathsf{ilab}[\mathsf{E}_{\mathcal{T}}[u]]|$. $|\mathsf{E}_{\mathcal{T}}|_{\mathsf{in}}$ is upper bounded by the total number of transitions $|\mathsf{E}_{\mathcal{T}}|$. Then, the following regret guarantee and computational complexity hold for FASTTRANSDUCE.

**Theorem 2.** *Let $(\mathcal{A}_{u,i})_{u \in Q, i \in \mathsf{ilab}[\mathsf{E}_{\mathcal{T}}[u]]}$ be external regret minimizing algorithms admitting data-dependent regret bounds of the form $O(\sqrt{L_T(\mathcal{A}_{u,i}) \log N})$, where $L_T(\mathcal{A}_{u,i})$ is the cumulative loss of $\mathcal{A}_{u,i}$ after $T$ rounds. Assume that, at each round, the sum of the minimal probabilities given to an expert by these algorithms is bounded below by some constant $\alpha > 0$. Then, FASTTRANSDUCE achieves a transductive regret against $\mathcal{T}$ that is in $O(\sqrt{T|\mathsf{E}_{\mathcal{T}}|_{\mathsf{in}} \log N})$ with a per-iteration complexity in $O\left(N^2 \min\left\{\frac{\log T}{\log(1/(1-\alpha))}, N\right\}\right)$.*

The proof is given in Appendix E. The regret guarantee of FASTTRANSDUCE matches that of the swap regret algorithm of Blum and Mansour [2007] or FASTSWAP in the case where $\mathcal{T}$ is chosen to be the family of swap transducers, and it matches the conditional $k$-gram swap regret of Mohri and Yang [2014] when $\mathcal{T}$ is chosen to be that of the $k$-gram swap transducers. Additionally, its computational complexity is typically more favorable than that of algorithms previously presented in the literature when the assumption on $\alpha$ holds, and it is never worse.

Remarkably, the computational complexity of FASTTRANSDUCE is comparable to the cost of FASTSWAP, even though FASTTRANSDUCE is a regret minimization algorithm against an arbitrary family of finite-state transducers. This is because only the external regret minimizing algorithms that correspond to the current state need to be updated at each round.

## 5  Time-selection transductive regret

In this section, we extend the notion of time-selection functions with modification rules to the setting of transductive regret and present an algorithm that achieves the same regret guarantee as Khot and Ponnuswami [2008] in their specific setting, but with a substantially more favorable computational complexity.

Time-selection functions were first introduced in [Lehrer, 2003] as boolean functions that determine which subset of times are relevant in the calculation of regret. This concept was relaxed to the real-valued setting by Blum and Mansour [2007] who considered time-selection functions taking values in $[0, 1]$. The authors introduced an algorithm which, for $K$ modification rules and $M$ time-selection functions, guarantees a regret in $O(\sqrt{TN \log(MK)})$ and admits a per-iteration complexity in $O(\max\{NKM, N^3\})$. For swap regret with time selection functions, this corresponds to a regret bound of $O(\sqrt{TN^2 \log(MN)})$ and a per-iteration computational cost in $O(N^{N+1}M)$. [Khot and Ponnuswami, 2008] improved upon this result and presented an algorithm with a regret bound in $O(\sqrt{T \log(MK)})$ and a per-iteration computational cost in $O(\max\{MK, N^3\})$, which is still prohibitively expensive for swap regret, since it is in $O(N^N M)$.

---
**Algorithm 2:** FASTTIMESELECTTRANSDUCE; $\mathcal{A}_\mathcal{I}$, $(\mathcal{A}_{I,u,i})$ external regret algorithms.
---
**Algorithm:** FASTTIMESELECTTRANSDUCE$(\mathcal{I}, \mathcal{T}, \mathcal{A}_\mathcal{I}, (\mathcal{A}_{I,u,i})_{I\in\mathcal{I},u\in Q_\mathcal{T},i\in\mathsf{ilab}[\mathsf{E}_\mathcal{T}[q]]})$

$u \leftarrow I_\mathcal{T}$
**for** $t \leftarrow 1$ **to** $T$ **do**
    **for each** $I \in \mathcal{I}$ **do**
        $\tilde{\mathsf{q}} \leftarrow \text{QUERY}(\mathcal{A}_\mathcal{I})$
        **for each** $i \in \mathsf{ilab}[\mathsf{E}_\mathcal{T}[u]]$ **do**
            $\mathsf{q}_{I,i} \leftarrow \text{QUERY}(\mathcal{A}_{I,u,i})$
        $\mathbf{M}^{t,u,I} \leftarrow [\mathsf{q}_{I,1}1_{1\in\mathsf{ilab}[\mathsf{E}_\mathcal{T}[u]]};\ldots;\mathsf{q}_{I,N}1_{N\in\mathsf{ilab}[\mathsf{E}_\mathcal{T}[u]]}]; \quad \mathbf{Q}^{t,u} \leftarrow \mathbf{Q}^{t,u} + I(t)\tilde{q}_I \mathbf{M}^{t,u,I};$
        $Z^t \leftarrow Z^t + I(t)\tilde{q}_I$
    $\mathbf{Q}^{t,u} \leftarrow \frac{\mathbf{Q}^{t,u}}{Z^t}$
    **for each** $j \leftarrow 1$ **to** $N$ **do**
        $c_j \leftarrow \min_{i\in\mathsf{ilab}[\mathsf{E}_\mathcal{T}[u]]} \mathbf{Q}^{t,u}_{i,j}1_{j\in\mathsf{ilab}[\mathsf{E}_\mathcal{T}[u]]}$
    $\alpha_t \leftarrow \|\mathbf{c}\|_1; \quad \tau_t \leftarrow \left\lceil \frac{\log\left(\frac{1}{\sqrt{t}}\right)}{\log(1-\alpha_t)} \right\rceil$
    **if** $\tau_t < N$ **then**
        $\mathsf{p}_t \leftarrow \mathsf{p}_t^0 \leftarrow \frac{\mathbf{c}}{\alpha_t}$
        **for** $\tau \leftarrow 1$ **to** $\tau_t$ **do**
            $(\mathsf{p}_t^\tau)^\top \leftarrow (\mathsf{p}_t^\tau)^\top(\mathbf{Q}^{t,u} - \vec{1}\mathbf{c}^\top); \quad \mathsf{p}_t \leftarrow \mathsf{p}_t + \mathsf{p}_t^\tau$
        $\mathsf{p}_t \leftarrow \frac{\mathsf{p}_t}{\|\mathsf{p}_t\|_1}$
    **else**
        $\mathsf{p}_t^\top \leftarrow \text{FIXED-POINT}(\mathbf{Q}^{t,u})$
    $x_t \leftarrow \text{SAMPLE}(\mathsf{p}_t); \quad \mathbf{l}_t \leftarrow \text{RECEIVELOSS}(); \quad u \leftarrow \delta_\mathcal{T}[u,x_t]$
    **for each** $I \in \mathcal{I}$ **do**
        $\tilde{l}_I^t \leftarrow I(t)\left(\mathsf{p}_t^\top \mathbf{M}^{t,u,I}\mathbf{l}_t - \mathsf{p}_t^\top \mathbf{l}_t\right)$
        **for each** $i \in \mathsf{ilab}[\mathsf{E}_\mathcal{T}[u]]$ **do**
            $\text{ATTRIBUTELOSS}(\mathcal{A}_{I,u,i}, \mathsf{p}_t[i]I(t)\mathbf{l}_t)$
    $\text{ATTRIBUTELOSS}(\mathcal{A}_\mathcal{I}, \tilde{\mathbf{l}}_t)$
---

We now formally define the scenario of online learning with time-selection transductive regret. Let $\mathcal{I} \subset [0,1]^\mathbb{N}$ be a family of time-selection functions. Each time-selection function $I \in \mathcal{I}$ determines the importance of the instantaneous regret at each round. Then, the *time-selection transductive regret* is defined as:

$$\text{Reg}_T(\mathcal{A}, \mathcal{I}, \Phi)$$
$$= \max_{I\in\mathcal{I}, \mathcal{T}\in\Phi} \left\{ \sum_{t=1}^{T} I(t) \underset{x_t\sim\mathsf{p}_t}{\mathbb{E}}[l_t(x_t)] - \sum_{t=1}^{T} I(t) \underset{x_t\sim\mathsf{p}_t}{\mathbb{E}} \left[ \sum_{e\in\mathsf{E}_\mathcal{T}[\delta_\mathcal{T}(I_\mathcal{T},x_{1:t-1}),x_t]} w[e]l_t(\mathsf{olab}[e]) \right] \right\}. \quad (3)$$

When the family of transducers admits a single state, this definition coincides with the notion of time-selection regret studied by Blum and Mansour [2007] or Khot and Ponnuswami [2008].

Time-selection transductive regret is a more difficult benchmark than transductive regret because the learner must account for only a subset of the rounds being relevant, in addition to playing a strategy that is robust against a large set of possible transductions.

To handle this scenario, we propose the following strategy. We maintain an external regret minimizing algorithm $\mathcal{A}_\mathcal{I}$ over the set of time-selection functions. This algorithm will be responsible for ensuring that our strategy is competitive against the a posteriori optimal time-selection function. We also maintain $|\mathcal{I}||Q|N$ other external regret minimizing algorithms, $\{\mathcal{A}_{I,u,i}\}_{I\in\mathcal{I},u\in Q_\mathcal{T},i\in\mathsf{ilab}[\mathsf{E}_\mathcal{T}[u]]}$, which will ensure that our algorithm is robust against each of the modification rules and the potential transductions. We will then use a meta-algorithm to assign appropriate surrogate losses to each of these external regret minimizing algorithms and combine them to form a stochastic matrix. As in FASTTRANSDUCE, this meta-algorithm will also approximate the stationary distribution of the matrix and use that as the learner's strategy. We call this algorithm FASTTIMESELECTTRANSDUCE. Its pseudocode is given in Algorithm 2.

**Theorem 3.** *Let* $(\mathcal{A}_{I,u,i})_{I\in\mathcal{I},u\in Q_{\mathcal{T}},i\in\mathsf{ilab}[\mathsf{E}_{\mathcal{T}}[q]]}$ *be external regret minimizing algorithms admitting data-dependent regret bounds of the form* $O(\sqrt{L_T(\mathcal{A}_{I,u,i})\log N})$*, where* $L_T(\mathcal{A}_{I,u,i})$ *is the cumulative loss of* $\mathcal{A}_{I,u,i}$ *after* $T$ *rounds. Let* $\mathcal{A}_{\mathcal{I}}$ *be an external regret minimizing algorithm over* $\mathcal{I}$ *that admits a regret in* $O(\sqrt{T\log(|\mathcal{I}|)})$ *after* $T$ *rounds. Assume further that at each round, the sum of the minimal probabilities given to an expert by these algorithms is bounded below by some constant* $\alpha > 0$*. Then,* FASTTIMESELECTTRANSDUCE *achieves a time-selection transductive regret with respect to the time-selection family* $\mathcal{I}$ *and WFST family* $\mathcal{T}$ *that is in* $O\Big(\sqrt{T\left(\log(|\mathcal{I}|)+|\mathsf{E}_{\mathcal{T}}|_{\mathsf{in}}\log N\right)}\Big)$ *with a per-iteration complexity in* $O\Big(N^2\Big(\min\Big\{\frac{\log(T)}{\log((1-\alpha)^{-1})},N\Big\}+|\mathcal{I}|\Big)\Big)$.

In particular, Theorem 3 implies that FASTTIMESELECTTRANSDUCE achieves the same time-selection swap regret guarantee as the algorithm of Khot and Ponnuswami [2008] but with a per-round computational cost that is only in $O\Big(N^2\Big(\min\Big\{\frac{\log(T)}{\log((1-\alpha)^{-1})},N\Big\}+|\mathcal{I}|\Big)\Big)$, as opposed to $O(|\mathcal{I}|N^N)$, which is an exponential improvement! Notice that this significant improvement does not require any assumption (it holds even for $\alpha = 0$).

## 6   Sleeping transductive regret

The standard setting of prediction with expert advice can be extended to the *sleeping experts* scenario studied by Freund et al. [1997], where, at each round, a subset of the experts are *asleep* and thus unavailable to the learner. The sleeping experts setting has been used to model problems appearing in text categorization [Cohen and Singer, 1999], calendar scheduling [Blum, 1997], or learning how to formulate search-engine queries [Cohen and Singer, 1996].

The standard benchmark in this setting is the *sleeping regret*, that is the difference between the cumulative expected loss of the learner and the cumulative expected loss of the best static distribution over the experts, restricted to and normalized over the set of awake experts $A_t \subseteq \Sigma$ at each round $t$:

$$\max_{\mathsf{u}\in\Delta_N}\left\{\sum_{t=1}^{T}\mathop{\mathbb{E}}_{x_t\sim\mathsf{p}_t^{A_t}}[l_t(x_t)]-\sum_{t=1}^{T}\mathop{\mathbb{E}}_{x_t\sim\mathsf{u}^{A_t}}[l_t(x_t)]\right\}. \tag{4}$$

Here, for any distribution $\mathsf{p}$, we use the notation $\mathsf{p}^{A_t}=\frac{\mathsf{p}|_{A_t}}{\sum_{i\in A_t}\mathsf{p}_i}$ with $\mathsf{p}|_A(a)=\mathsf{p}(a)1_{a\in A}$, for any $a\in\Sigma$ and $A\subseteq\Sigma$. An alternative definition of sleeping regret studied and bounded by Freund et al. [1997] is the following:

$$\max_{\mathsf{u}\in\Delta_N}\left\{\sum_{t=1}^{T}\mathsf{u}(A_t)\mathop{\mathbb{E}}_{x_t\sim\mathsf{p}_t^{A_t}}[l_t(x_t)]-\sum_{t=1}^{T}\mathop{\mathbb{E}}_{x_t\sim\mathsf{u}}[1_{x_t\in A_t}l_t(x_t)]\right\}. \tag{5}$$

This is also the definition we will be adopting in our analysis. Note that if $\mathsf{u}(A_t)$ does not vary with $t$, then the two definitions only differ by a multiplicative constant. By generalizing the results of Freund et al. [1997] to arbitrary losses, that is beyond those that satisfy equation (6) in their paper, one can show that there exist algorithms with sleeping regret in $O\Big(\sqrt{\sum_{t=1}^{T}\mathsf{u}^*(A_t)\mathbb{E}_{x_t\sim\mathsf{p}_t}[l_t(x_t)]\log(N)}\Big)$, where $\mathsf{u}^*$ maximizes the expression to be bounded.

In this section, we extend this definition of sleeping regret to *sleeping transductive regret*, that is the difference between the learner's cumulative expected loss and the cumulative expected loss of any transduction of the learner's actions among a family of finite-state transducers, where the weights of the transductions are normalized over the set of awake experts. The sleeping transductive regret can be expressed as follows:

$$\mathrm{Reg}_T(\mathcal{A},\mathcal{T},A_1^T)=\max_{\substack{\tau\in\mathcal{T}\\\mathsf{u}\in\Delta_N}}\left\{\sum_{t=1}^{T}\mathsf{u}(A_t)\mathop{\mathbb{E}}_{x_t\sim\mathsf{p}_t^{A_t}}[l_t(x_t)]\right.$$
$$\left.-\sum_{t=1}^{T}\mathop{\mathbb{E}}_{x_t\sim\mathsf{p}_t^{A_t}}\left[\sum_{e\in\mathsf{E}_{\mathcal{T}}[\delta_{\mathcal{T}}(I_{\mathcal{T}},x_{1:t-1}),x_t]}(\mathsf{u}|_{A_t})_{\mathsf{olab}[e]}w[e]l_t(\mathsf{olab}[e])\right]\right\}. \tag{6}$$

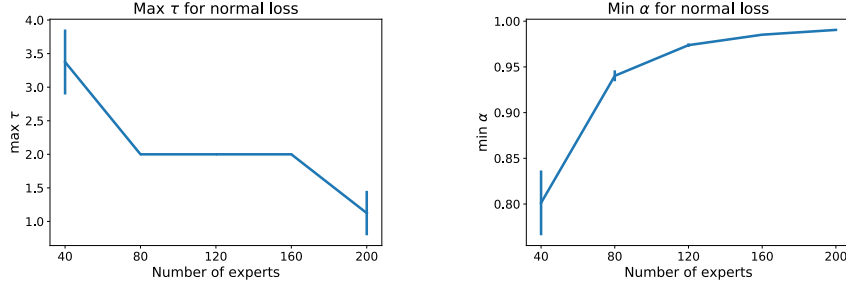

Figure 3: Maximum values of $\tau$ and minimum values of $\alpha$ in FASTSWAP experiments. The vertical bars represent the standard deviation across $16$ instantiations of the same simulation.

When all experts are awake at every round, i.e. $A_t = \Sigma$, the sleeping transductive regret reduces to the standard transductive regret. When the family of transducers corresponds to that of swap regret, we uncover a natural definition for sleeping swap regret: $\max_{\varphi \in \Phi_{\text{swap}}, \mathsf{u} \in \Delta_N} \sum_{t=1}^{T} \mathsf{u}(A_t) \mathbb{E}_{x_t \sim \mathsf{p}_t^{A_t}} [l_t(x_t)] - \sum_{t=1}^{T} \mathbb{E}_{x_t \sim \mathsf{p}_t^{A_t}} \left[ \mathsf{u}_{\varphi(x_t)} 1_{\varphi(x_t) \in A_t} l_t(\varphi(x_t)) \right]$. We now present an efficient algorithm for minimizing sleeping transductive regret, FASTSLEEPTRANSDUCE. Similar to FASTTRANSDUCE, this algorithm uses a meta-algorithm with multiple regret minimizing sub-algorithms and a fixed-point approximation to compute the learner's strategy. However, since FASTSLEEPTRANSDUCE minimizes sleeping transductive regret, it uses sleeping regret minimizing sub-algorithms (i.e. those with regret guarantees of the form (5)). The meta-algorithm also designs a different stochastic matrix. The pseudocode of this algorithm is given in Appendix C.

**Theorem 4.** *Assume that the sleeping regret minimizing algorithms used as inputs of* FASTSLEEPTRANSDUCE *achieve data-dependent regret bounds such that, if the algorithm selects the distributions $(\mathsf{p}_t)_{t=1}^{T}$ and observes losses $(\mathsf{l}_t)_{t=1}^{T}$ with awake sets $(A_t)_{t=1}^{T}$, then the regret of $\mathcal{A}_i^q$ is at most $O\!\left( \sqrt{\sum_{t=1}^{T} \mathsf{u}^*(A_t) \mathbb{E}_{x_t \sim \mathsf{p}_t} [l_t(x_t)] \log(N)} \right)$. Assume further that at each round, the sum of the minimal probabilities given to an expert by these algorithms is bounded below by some constant $\alpha > 0$. Then, the sleeping regret $\mathrm{Reg}_T(\text{FASTSLEEPTRANSDUCE}, \mathcal{T}, A_1^T)$ of* FASTSLEEPTRANSDUCE *is upper bounded by $O\!\left( \sqrt{\sum_{t=1}^{T} \mathsf{u}(A_t) |\mathsf{E}_{\mathcal{T}}|_{\text{in}} \log(N)} \right)$. Moreover,* FASTSLEEPTRANSDUCE *admits a per-iteration complexity in $O\!\left( N^2 \min\left\{ \frac{\log T}{\log(1/(1-\alpha))}, N \right\} \right)$.*

# 7  Experiments

In this section, we present some toy experiments illustrating the effectiveness of the Reduced Power Method for approximating the stationary distribution in FASTSWAP.

We considered $n$ base learners, where $n \in \{40, 80, 120, 160, 200\}$, each using the weighted-majority algorithm [Littlestone and Warmuth, 1994]. We generated losses as i.i.d. normal random variables with means in $(0.1, 0.9)$ (chosen randomly) and standard deviation equal to $0.1$. We capped the losses above and below to remain in $[0, 1]$. We ran FASTSWAP for $10{,}000$ rounds in each simulation and repeated each simulation $16$ times. The plot of the maximum $\tau$ for each simulation is shown in Figure 3. Across all simulations, the maximum $\tau$ attained was $4$, so that at most $4$ iterations of the RPM were needed on any given round to obtain a sufficient approximation. Thus, the per-iteration cost in these simulations was indeed in $\widetilde{O}(N^2)$, an improvement over the $O(N^3)$ cost in prior work.

# 8  Conclusion

We introduced the notion of transductive regret, further extended it to the time-selection and sleeping experts settings, and presented efficient online learning algorithms for all these setting with sublinear transductive regret guarantees. We both generalized the existing theory and gave more efficient algorithms in existing subcases. The algorithms and results in this paper can be further extended to the case of fully non-deterministic weighted finite-state transducers.

**Acknowledgments**

We thank Avrim Blum for informing us of an existing lower bound for swap regret proven by Auer [2017]. This work was partly funded by NSF CCF-1535987 and NSF IIS-1618662.

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
