[Supplementary Material]

# A  Additional figures and examples

## A.1  Special cases of transductive regret.

Figure 4: Several families of WFSTs for special cases of transductive regret for $\Sigma = \{a, b, c\}$. (i) External regret with parameter $x \in \Sigma$. (ii) Internal regret: family of transducers $\mathcal{T}_{a_1, a_2}$ with $a_1 \neq a_2$, $a_1, a_2 \in \Sigma$; example shown for $\mathcal{T}_{a,b}$. (iii) Swap regret with parameter $\varphi \colon \Sigma \to \Sigma$. (iv) Bigram conditional swap regret with parameter $\psi \colon (\Sigma \cup \{\epsilon\}) \times \Sigma \to \Sigma$.

## A.2 Example with a swapping subset.

Figure 5: Example of a WFST with $\Sigma = \{a, b, c, d\}$ and where each state has a swapping subset.

# B    Pseudocode of FASTTRANSDUCE

---

**Algorithm 3:** FASTTRANSDUCE; $(\mathcal{A}_{u,i})_{u \in Q_{\mathcal{T}}, i \in \mathsf{ilab}[\mathsf{E}_{\mathcal{T}}[u]]}$ external regret minimization algorithms.

---

**Algorithm:** FASTTRANSDUCE$(\mathcal{T}, (\mathcal{A}_{u,i})_{u \in Q_{\mathcal{T}}, i \in \mathsf{ilab}[\mathsf{E}_{\mathcal{T}}[u]]})$

$u \leftarrow I_{\mathcal{T}}$
**for** $t \leftarrow 1$ **to** $T$ **do**
    **for each** $i \in \mathsf{ilab}[\mathsf{E}_{\mathcal{T}}[u]]$ **do**
        $\mathsf{q}_i \leftarrow \text{QUERY}(\mathcal{A}_{u,i})$
    $\mathbf{Q}^{t,u} \leftarrow [\mathsf{q}_1 1_{1 \in \mathsf{ilab}[\mathsf{E}_{\mathcal{T}}[u]]} \cdots \mathsf{q}_N 1_{N \in \mathsf{ilab}[\mathsf{E}_{\mathcal{T}}[u]]}]^{\top}$
    **for each** $j \leftarrow 1$ **to** $N$ **do**
        $c_j \leftarrow \min_{i \in \mathsf{ilab}[\mathsf{E}_{\mathcal{T}}[u]]} \mathbf{Q}^{t,u}_{i,j} 1_{j \in \mathsf{ilab}[\mathsf{E}_{\mathcal{T}}[u]]}$
    $\alpha_t \leftarrow \|\mathbf{c}\|_1; \quad \tau_t \leftarrow \left\lceil \frac{\log\left(\frac{1}{\sqrt{t}}\right)}{\log(1 - \alpha_t)} \right\rceil$
    **if** $\tau_t < N$ **then**
        $\mathsf{p}_t \leftarrow \mathsf{p}_t^0 \leftarrow \frac{\mathbf{c}}{\alpha_t}$
        **for** $\tau \leftarrow 1$ **to** $\tau_t$ **do**
            $(\mathsf{p}_t^{\tau})^{\top} \leftarrow (\mathsf{p}_t^{\tau})^{\top}(\mathbf{Q}^{t,u} - \vec{1}\mathbf{c}^{\top}); \mathsf{p}_t \leftarrow \mathsf{p}_t + \mathsf{p}_t^{\tau}$
        $\mathsf{p}_t \leftarrow \frac{\mathsf{p}_t}{\|\mathsf{p}_t\|_1}$
    **else**
        $\mathsf{p}_t^{\top} = \text{FIXED-POINT}(\mathbf{Q}^{t,u})$
    $x_t \leftarrow \text{SAMPLE}(\mathsf{p}_t); \quad \mathbf{l}_t \leftarrow \text{RECEIVELOSS}(); \quad u \leftarrow \delta_{\mathcal{T}}(u, x_t)$
    **for each** $i \in \mathsf{ilab}[\mathsf{E}_{\mathcal{T}}[u]]$ **do**
        $\text{ATTRIBUTELOSS}(\mathcal{A}_{u,i}, \mathsf{p}_t[i]\mathbf{l}_t)$

---

# C   Pseudocode of FASTSLEEPTRANSDUCE

---

**Algorithm 4:** FASTSLEEPTRANSDUCE. $(\mathcal{A}_{u,i})$ sleeping regret minimization algorithms.

---

**Algorithm:** FASTSLEEPTRANSDUCE$(\mathcal{T}, \{\mathcal{A}_{u,i}\}_{u\in Q_{\mathcal{T}}, i\in \mathsf{ilab}[\mathsf{E}_{\mathcal{T}}[u]]})$

$u \leftarrow I_{\mathcal{T}}$
**for** $t \leftarrow 1$ **to** $T$ **do**
    $A_t \leftarrow \text{AWAKESET}()$
    **for each** $i \in \mathsf{ilab}[\mathsf{E}_{\mathcal{T}}[u]] \cap A_t$ **do**
        $\mathsf{q}_i \leftarrow \text{QUERY}(\mathcal{A}_{u,i}); \quad \mathsf{q}_i^{A_t} \leftarrow \frac{\mathsf{q}_i|_{A_t}}{\sum_{j\in A_t} \mathsf{q}_i}$
    $\mathbf{Q}^{t,u} \leftarrow [\mathsf{q}_1^{A_t} 1_{1\in \mathsf{ilab}[\mathsf{E}_{\mathcal{T}}[u]]\cap A_t}; \ldots; \mathsf{q}_N^{A_t} 1_{N\in \mathsf{ilab}[\mathsf{E}_{\mathcal{T}}[u]]\cap A_t}]$
    **for each** $j \leftarrow 1$ **to** $N$ **do**
        $c_j \leftarrow \min_{i\in \mathsf{ilab}[\mathsf{E}_{\mathcal{T}}[u]]\cap A_t} \mathbf{Q}_{i,j}^{t,u} 1_{j\in \mathsf{ilab}[\mathsf{E}_{\mathcal{T}}[u]]\cap A_t}$
    $\alpha_t \leftarrow \|\mathbf{c}\|_1; \quad \tau_t \leftarrow \left\lceil \frac{\log\left(\frac{1}{\sqrt{t}}\right)}{\log(1-\alpha_t)} \right\rceil$
    **if** $\tau_t < N$ **then**
        $\mathsf{p}_t \leftarrow \mathsf{p}_t^0 \leftarrow \frac{\mathbf{c}}{\alpha_t}$
        **for** $\tau \leftarrow 1$ **to** $\tau_t$ **do**
            $(\mathsf{p}_t^\tau)^\top \leftarrow (\mathsf{p}_t^\tau)^\top (\mathbf{Q}^{t,u} - [1_{1\in A_t}; \ldots; 1_{|\mathsf{ilab}[\mathsf{E}_{\mathcal{T}}[q]]|\in A_t}]\mathbf{c}^\top)$
            $\mathsf{p}_t \leftarrow \mathsf{p}_t + \mathsf{p}_t^\tau$
        $\mathsf{p}_t \leftarrow \frac{\mathsf{p}_t}{\|\mathsf{p}_t\|_1}$
    **else**
        $\mathsf{p}_t^\top \leftarrow \text{FIXED-POINT}(\mathbf{Q}^{t,u})$
    $\mathsf{p}_t^{A_t} \leftarrow \frac{\mathsf{p}_t|_{A_t}}{\sum_{j\in A_t}\mathsf{p}_{t,j}}; \quad x_t \leftarrow \text{SAMPLE}(\mathsf{p}_t^{A_t}); \quad \mathbf{l}_t \leftarrow \text{RECEIVELOSS}(); \quad u \leftarrow \delta_{\mathcal{T}}[u, x_t]$
    **for each** $i \in \mathsf{ilab}[\mathsf{E}_{\mathcal{T}}[u]]$ **do**
        $\text{ATTRIBUTELOSS}(\mathcal{A}_{u,i}, \mathsf{p}_t[i]\mathbf{l}_t)$

---

## D  Proof of Theorem 1

**Theorem 1.** *Let $\mathcal{A}_1, \ldots, \mathcal{A}_N$ be external regret minimizing algorithms admitting data-dependent regret bounds of the form $O(\sqrt{L_T(\mathcal{A}_i) \log N})$, where $L_T(\mathcal{A}_i)$ is the cumulative loss of $\mathcal{A}_i$ after $T$ rounds. Assume that, at each round, the sum of the minimal probabilities given to an expert by these algorithms is bounded below by some constant $\alpha > 0$. Then,* FASTSWAP *achieves a swap regret in $O(\sqrt{TN \log N})$ with a per-iteration complexity in $O\left(N^2 \min\left\{\frac{\log T}{\log(1/(1-\alpha))}, N\right\}\right)$.*

*Proof.* Let $\mathsf{p}_t$ be the distribution returned by FASTSWAP at round $t$. For any distribution $\mathsf{p}_t^*$, $t \in [T]$, the following inequality holds:

$$
\sum_{t=1}^{T} \underset{x_t \sim \mathsf{p}_t}{\mathbb{E}}[l_t(x_t)] 1_{\tau_t < N} = \sum_{t=1}^{T} \underset{x_t \sim \mathsf{p}_t^*}{\mathbb{E}}[l_t(x_t)] 1_{\tau_t < N} + \sum_{t=1}^{T} \underset{x_t \sim \mathsf{p}_t}{\mathbb{E}}[l_t(x_t)] 1_{\tau_t < N}
$$

$$
- \sum_{t=1}^{T} \underset{x_t \sim \mathsf{p}_t^*}{\mathbb{E}}[l_t(x_t)] 1_{\tau_t < N}
$$

$$
\leq \sum_{t=1}^{T} \underset{x_t \sim \mathsf{p}_t^*}{\mathbb{E}}[l_t(x_t)] 1_{\tau_t < N} + \sum_{t=1}^{T} \|\mathsf{p}_t - \mathsf{p}_t^*\|_1 \|l_t\|_\infty 1_{\tau_t < N}
$$

$$
\leq \sum_{t=1}^{T} \underset{x_t \sim \mathsf{p}_t^*}{\mathbb{E}}[l_t(x_t)] 1_{\tau_t < N} + \sum_{t=1}^{T} \|\mathsf{p}_t - \mathsf{p}_t^*\|_1 1_{\tau_t < N}.
$$

Let $\mathsf{p}_t^*$ be the stationary distribution of the row stochastic matrix $\mathbf{Q}^t$, $\mathsf{p}_t^{*\top} \mathbf{Q}^t = \mathsf{p}_t^{*\top}$. Then, we can write

$$
\sum_{t=1}^{T} \underset{x_t \sim \mathsf{p}_t^*}{\mathbb{E}}[l_t(x_t)] 1_{\tau_t < N} = \sum_{t=1}^{T} \sum_{j=1}^{N} \mathsf{p}_{t,j}^* l_{t,j} 1_{\tau_t < N}
$$

$$
= \sum_{t=1}^{T} \sum_{i=1}^{N} \sum_{j=1}^{N} \mathsf{p}_{t,i}^* \mathbf{Q}_{i,j}^t l_{t,j} 1_{\tau_t < N}
$$

$$
= \sum_{i=1}^{N} \sum_{t=1}^{T} \sum_{j=1}^{N} \mathbf{Q}_{i,j}^t \mathsf{p}_{t,i} l_{t,j} 1_{\tau_t < N} + \sum_{i=1}^{N} \sum_{t=1}^{T} \sum_{j=1}^{N} \mathbf{Q}_{i,j}^t (\mathsf{p}_{t,i}^*
$$

$$
- \mathsf{p}_{t,i}) l_{t,j} 1_{\tau_t < N}
$$

$$
\leq \sum_{i=1}^{N} \sum_{t=1}^{T} \sum_{j=1}^{N} \mathbf{Q}_{i,j}^t \mathsf{p}_{t,i} l_{t,j} 1_{\tau_t < N} + \sum_{t=1}^{T} \|\mathsf{p}_t^* - \mathsf{p}_t\|_1 1_{\tau_t < N}.
$$

On the other hand, by design, if $\tau_t \geq N$, then $\mathsf{p}_t = \mathsf{p}_t^*$, so that

$$
\sum_{t=1}^{T} \underset{x_t \sim \mathsf{p}_t}{\mathbb{E}}[l_t(x_t)] 1_{\tau_t \geq N} = \sum_{i=1}^{N} \sum_{t=1}^{T} \sum_{j=1}^{N} \mathbf{Q}_{i,j}^t \mathsf{p}_{t,i} l_{t,j} 1_{\tau_t \geq N}.
$$

Thus, it follows that

$$
\sum_{t=1}^{T} \underset{x_t \sim \mathsf{p}_t}{\mathbb{E}}[l_t(x_t)] \leq \sum_{i=1}^{N} \sum_{t=1}^{T} \sum_{j=1}^{N} \mathbf{Q}_{i,j}^t \mathsf{p}_{t,i} l_{t,j} + 2 \sum_{t=1}^{T} \|\mathsf{p}_t^* - \mathsf{p}_t\|_1 1_{\tau_t < N}
$$

$$
\leq \sum_{i=1}^{N} \left[ \min_{j \in [N]} \sum_{t=1}^{T} \mathsf{p}_{t,i} l_{t,j} + \mathrm{Reg}_T(\mathcal{A}_i, \Phi_{\text{ext}}) \right] + 2 \sum_{t=1}^{T} \|\mathsf{p}_t^* - \mathsf{p}_t\|_1 1_{\tau_t < N}
$$

$$
= \min_{\varphi \in \Phi_{\text{swap}}} \sum_{i=1}^{N} \left[ \sum_{t=1}^{T} \mathsf{p}_{t,i} l_{t,\varphi(i)} + \mathrm{Reg}_T(\mathcal{A}_i, \Phi_{\text{ext}}) \right] + 2 \sum_{t=1}^{T} \|\mathsf{p}_t^* - \mathsf{p}_t\|_1 1_{\tau_t < N}.
$$

Now let $L_T(\mathcal{A}_i)$ denote the cumulative loss incurred by algorithm $\mathcal{A}_i$. Since the losses attributed to algorithm $\mathcal{A}_i$ are scaled by $\mathsf{p}_{t,i}$, at each round, the sum of the losses over all the algorithms is at most 1. Thus, by Jensen's inequality, the following inequalities hold:

$$\frac{1}{N}\sum_{i=1}^{N}\mathrm{Reg}_T(\mathcal{A}_i,\Phi_{\mathrm{ext}}) = \frac{1}{N}\sum_{i=1}^{N}O\left(\sqrt{L_T(\mathcal{A}_i)\log N}\right)$$

$$\leq O\left(\sqrt{\frac{1}{N}\sum_{i=1}^{N}L_T(\mathcal{A}_i)\log N}\right) \leq O\left(\sqrt{\frac{T\log N}{N}}\right),$$

which implies $\sum_{i=1}^{N}\mathrm{Reg}_T(\mathcal{A}_i,\Phi_{\mathrm{ext}}) \leq \sqrt{TN\log N}$.

Finally, during the rounds in which $1_{\tau_t<N}$, $\mathsf{p}_t$ is an RPM approximation of $\mathsf{p}_t^*$ using $\tau_t$ iterations. Thus, by Equation 3.7 in [Nesterov and Nemirovski, 2015] the following inequality holds: $\|\mathsf{p}_t - \mathsf{p}_t^*\|_1 \leq (1-\alpha_t)^{\tau_t}$. Since $\tau_t$ is chosen so that the inequality $(1-\alpha_t)^{\tau_t} \leq 1/\sqrt{t}$ holds, it follows that $\sum_{t=1}^{T}\|\mathsf{p}_t - \mathsf{p}_t^*\|1_{\tau_t<N} \leq \sum_{t=1}^{T}1/\sqrt{t} \leq \sqrt{T}$, which proves the regret bound $\mathrm{Reg}_T(\mathcal{A},\Phi_{\mathrm{swap}}) \leq O(\sqrt{TN\log N})$.

Furthermore, the computational cost of the $t$-th iteration of the algorithm is dominated by $\tau_t$ matrix multiplications or the solution of the linear system. $\tau_t$ can be bounded as follows: $\tau_t = \left\lceil \frac{\log\left(\frac{1}{\sqrt{t}}\right)}{\log(1-\alpha_t)} \right\rceil \leq \frac{\log\left(\frac{1}{\sqrt{t}}\right)}{\log(1-\alpha)} + 1$. Thus, the computational cost of the $t$-th iteration is in

$$O\left(N^2\min\left\{\frac{\log t}{\log(1/(1-\alpha_t))}, N\right\}\right) \leq O\left(N^2\min\left\{\frac{\log T}{\log(1/(1-\alpha))}, N\right\}\right).$$

$\square$

# E   Proof of Theorem 2

**Theorem 2.** *Let $(\mathcal{A}_{u,i})_{u \in Q, i \in \mathsf{ilab}[\mathsf{E}_{\mathcal{T}}[u]]}$ be external regret minimizing algorithms admitting data-dependent regret bounds of the form $O(\sqrt{L_T(\mathcal{A}_{u,i}) \log N})$, where $L_T(\mathcal{A}_{u,i})$ is the cumulative loss of $\mathcal{A}_{u,i}$ after $T$ rounds. Assume that, at each round, the sum of the minimal probabilities given to an expert by these algorithms is bounded below by some constant $\alpha > 0$. Then, FASTTRANSDUCE achieves a transductive regret against $\mathcal{T}$ that is in $O(\sqrt{T|\mathsf{E}_{\mathcal{T}}|_{\mathsf{in}} \log N})$ with a per-iteration complexity in $O\left(N^2 \min\left\{\frac{\log T}{\log(1/(1-\alpha))}, N\right\}\right)$.*

*Proof.* Let $\mathsf{p}_t$ be the distribution output by FASTTRANSDUCE at round $t$. For any distribution $\mathsf{p}_t^*$, $t \in [T]$, the following inequalities hold:

$$
\begin{aligned}
\sum_{t=1}^{T} \mathbb{E}_{x_t \sim \mathsf{p}_t}[l_t(x_t)]\mathbf{1}_{\tau_t < N} = &\sum_{t=1}^{T} \mathbb{E}_{x_t \sim \mathsf{p}_t^*}[l_t(x_t)]\mathbf{1}_{\tau_t < N} + \sum_{t=1}^{T} \mathbb{E}_{x_t \sim \mathsf{p}_t}[l_t(x_t)]\mathbf{1}_{\tau_t < N} \\
&- \sum_{t=1}^{T} \mathbb{E}_{x_t \sim \mathsf{p}_t^*}[l_t(x_t)]\mathbf{1}_{\tau_t < N} \\
\leq &\sum_{t=1}^{T} \mathbb{E}_{x_t \sim \mathsf{p}_t^*}[l_t(x_t)]\mathbf{1}_{\tau_t < N} + \sum_{t=1}^{T} \|\mathsf{p}_t - \mathsf{p}_t^*\|_1 \|l_t\|_\infty \mathbf{1}_{\tau_t < N} \\
\leq &\sum_{t=1}^{T} \mathbb{E}_{x_t \sim \mathsf{p}_t^*}[l_t(x_t)]\mathbf{1}_{\tau_t < N} + \sum_{t=1}^{T} \|\mathsf{p}_t - \mathsf{p}_t^*\|_1 \mathbf{1}_{\tau_t < N}.
\end{aligned}
$$

Let $u_t$ be the state that the algorithm is in at time $t$ as a result of its past actions. Consider the matrix $\mathbf{Q}^{t,u_t}$ defined in the algorithm. The restriction of the matrix $\mathbf{Q}^{t,u_t}$ to its non-zero rows and columns is a row stochastic matrix. Let $\mathsf{p}_t^*$ be its stationary distribution, and by augmenting it with zeros in the zero rows of $\mathbf{Q}^{t,u_t}$, we may take $\mathsf{p}_t^* \in \Delta_N$ as a fixed point of $\mathbf{Q}^{t,u_t}$. Then, we can write:

$$
\begin{aligned}
\sum_{t=1}^{T} \mathbb{E}_{x_t \sim \mathsf{p}_t^*}[l_t(x_t)]\mathbf{1}_{\tau_t < N} = &\sum_{t=1}^{T} \sum_{i=1}^{N} \sum_{j=1}^{N} \mathsf{p}_{t,i}^* \mathbf{Q}_{i,j}^{t,u_t} l_{t,j} \mathbf{1}_{\tau_t < N} \\
= &\sum_{i=1}^{N} \sum_{t=1}^{T} \sum_{j=1}^{N} \mathbf{Q}_{i,j}^{t,u_t} \mathsf{p}_{t,i} l_{t,j} \mathbf{1}_{\tau_t < N} \\
&+ \sum_{i=1}^{N} \sum_{t=1}^{T} \sum_{j=1}^{N} \mathbf{Q}_{i,j}^{t,u_t} (\mathsf{p}_{t,i}^* - \mathsf{p}_{t,i}) l_{t,j} \mathbf{1}_{\tau_t < N} \\
\leq &\sum_{i=1}^{N} \sum_{t=1}^{T} \sum_{j=1}^{N} \mathbf{Q}_{i,j}^{t,u_t} \mathsf{p}_{t,i} l_{t,j} \mathbf{1}_{\tau_t < N} + \sum_{t=1}^{T} \|\mathsf{p}_t^* - \mathsf{p}_t\|_1 \mathbf{1}_{\tau_t < N}.
\end{aligned}
$$

On the other hand, by design, if $\tau_t \geq N$, then $\mathsf{p}_t = \mathsf{p}_t^*$, so that

$$
\sum_{t=1}^{T} \mathbb{E}_{x_t \sim \mathsf{p}_t}[l_t(x_t)]\mathbf{1}_{\tau_t \geq N} = \sum_{i=1}^{N} \sum_{t=1}^{T} \sum_{j=1}^{N} \mathbf{Q}_{i,j}^{t,u_t} \mathsf{p}_{t,i} l_{t,j} \mathbf{1}_{\tau_t \geq N}.
$$

Thus, it follows that for any WFST $\mathcal{T} \in \mathcal{T}$,

$$
\begin{aligned}
\sum_{t=1}^{T} \mathbb{E}_{x_t \sim \mathsf{p}_t}[l_t(x_t)] \leq &\sum_{i=1}^{N} \sum_{t=1}^{T} \sum_{j=1}^{N} \sum_{u \in Q_{\mathcal{T}}} \mathbf{Q}_{i,j}^{t,u} \mathbf{1}_{\delta_{\mathcal{T}}(I_{\mathcal{T}}, x_{1:t-1})=u} \mathsf{p}_{t,i} l_{t,j} + 2\sum_{t=1}^{T} \|\mathsf{p}_t^* - \mathsf{p}_t\|_1 \mathbf{1}_{\tau_t < N} \\
= &\sum_{u \in Q_{\mathcal{T}}} \sum_{i \in \mathsf{ilab}[\mathsf{E}_{\mathcal{T}}[u]]} \sum_{t=1}^{T} \sum_{j=1}^{N} \mathbf{Q}_{i,j}^{t,u} \mathbf{1}_{\delta_{\mathcal{T}}(I_{\mathcal{T}}, x_{1:t-1})=u} \mathsf{p}_{t,i} l_{t,j}
\end{aligned}
$$

$$+ 2\sum_{t=1}^{T} \|\mathsf{p}_t^* - \mathsf{p}_t\|_1 1_{\tau_t < N}$$

$$\leq \sum_{u \in Q_{\mathcal{T}}} \sum_{i \in \mathsf{ilab}[\mathsf{E}_{\mathcal{T}}[u]]} \min_{i^* \in \mathsf{olab}[\mathsf{E}_{\mathcal{T}}[\delta_{\mathcal{T}}(I_{\mathcal{T}}, x_{1:t-1}), x_t]]} \sum_{t=1}^{T} 1_{\delta_{\mathcal{T}}(I_{\mathcal{T}}, x_{1:t-1})=u} \mathsf{p}_{t,i} l_{t,i^*}$$

$$+ 2\sum_{t=1}^{T} \|\mathsf{p}_t^* - \mathsf{p}_t\|_1 1_{\tau_t < N} + \sum_{i=1}^{N} \sum_{u \in Q_{\mathcal{T}}} \mathrm{Reg}_T(\mathcal{A}_{u,i}, \Phi_{\mathrm{ext}})$$

$$\leq \sum_{u \in Q_{\mathcal{T}}} \sum_{i \in \mathsf{ilab}[\mathsf{E}_{\mathcal{T}}[q]]} \sum_{e \in \mathsf{E}_{\mathcal{J}}[\delta_{\mathcal{T}}(I_{\mathcal{T}}, x_{1:t-1}), x_t]} \sum_{t=1}^{T} 1_{\delta_{\mathcal{T}}(I_{\mathcal{T}}, x_{1:t-1})=u} \mathsf{p}_{t,i} w[e] l_t(\mathsf{olab}[e])$$

$$+ 2\sum_{t=1}^{T} \|\mathsf{p}_t^* - \mathsf{p}_t\|_1 1_{\tau_t < N} + \sum_{u \in Q_{\mathcal{T}}} \sum_{i \in \mathsf{ilab}[\mathsf{E}_{\mathcal{T}}[q]]} \mathrm{Reg}_T(\mathcal{A}_{u,i}, \Phi_{\mathrm{ext}})$$

$$= \sum_{t=1}^{T} \mathbb{E}_{x_t \sim \mathsf{p}_t} \left[ \sum_{e \in \mathsf{E}_{\mathcal{J}}[\delta_{\mathcal{T}}(I_{\mathcal{T}}, x_{1:t-1}), x_t]} w[e] l_t(\mathsf{olab}[e]) \right] + 2\sum_{t=1}^{T} \|\mathsf{p}_t^* - \mathsf{p}_t\|_1 1_{\tau_t < N}$$

$$+ \sum_{u \in Q_{\mathcal{T}}} \sum_{i \in \mathsf{ilab}[\mathsf{E}_{\mathcal{T}}[q]]} \mathrm{Reg}_T(\mathcal{A}_{u,i}, \Phi_{\mathrm{ext}}).$$

Now let $L_T(\mathcal{A}_{u,i})$ denote the cumulative loss incurred by algorithm $\mathcal{A}_{u,i}$. Since the losses attributed to algorithm $\mathcal{A}_{u,i}$ are scaled by $1_{\delta_{\mathcal{T}}(I_{\mathcal{T}}, x_{1:t-1})=u} \mathsf{p}_{t,i}$, it follows that at each round, the sum of the losses over all the algorithms is at most 1. Thus, by Jensen's inequality, it follows that

$$\frac{1}{\sum_{u \in Q_{\mathcal{T}}} |\mathsf{ilab}[\mathsf{E}_{\mathcal{T}}[u]]|} \sum_{u \in Q_{\mathcal{T}}} \sum_{i \in \mathsf{ilab}[\mathsf{E}_{\mathcal{T}}[u]]} \mathrm{Reg}_T(\mathcal{A}_{u,i}, \Phi_{\mathrm{ext}})$$

$$= \frac{1}{\sum_{u \in Q_{\mathcal{T}}} |\mathsf{ilab}[\mathsf{E}_{\mathcal{T}}[u]]|} \sum_{u \in Q_{\mathcal{T}}} \sum_{i \in \mathsf{ilab}[\mathsf{E}_{\mathcal{T}}[u]]} \sqrt{L_T(\mathcal{A}_{u,i}) \log(N)}$$

$$\leq \sqrt{\frac{1}{\sum_{u \in Q_{\mathcal{T}}} |\mathsf{ilab}[\mathsf{E}_{\mathcal{T}}[u]]|} \sum_{u \in Q_{\mathcal{T}}} \sum_{i \in \mathsf{ilab}[\mathsf{E}_{\mathcal{J}}[u]]} L_T(\mathcal{A}_{u,i}) \log(N)}$$

$$\leq \sqrt{\frac{1}{\sum_{u \in Q_{\mathcal{T}}} |\mathsf{ilab}[\mathsf{E}_{\mathcal{T}}[u]]|} T \log(N)},$$

so that $\sum_{u \in Q_{\mathcal{T}}} \sum_{i \in \mathsf{ilab}[\mathsf{E}_{\mathcal{T}}[u]]} \mathrm{Reg}_T(\mathcal{A}_{u,i}, \Phi_{\mathrm{ext}}) \leq \sqrt{T \sum_{u \in Q_{\mathcal{T}}} |\mathsf{ilab}[\mathsf{E}_{\mathcal{T}}[u]]| \log(N)}$.

Finally, during the rounds in which $1_{\tau_t < N}$, $\mathsf{p}_t$ is an RPM approximation of $\mathsf{p}_t^*$ using $\tau_t$ iterations. Thus, it follows from Equation 3.7 in [Nesterov and Nemirovski, 2015] that $\|\mathsf{p}_t - \mathsf{p}_t^*\|_1 \leq (1 - \alpha_t)^{\tau_t}$. By the algorithm's choice of $\tau_t$, $\|\mathsf{p}_t - \mathsf{p}_t^*\|_1 \leq \frac{1}{\sqrt{t}}$. Thus, it follows that $\sum_{t=1}^{T} \|\mathsf{p}_t - \mathsf{p}_t^*\|_1 1_{\tau_t < N} \leq \sqrt{T}$, so that $\mathrm{Reg}_T(\mathcal{A}, \mathcal{T}) \leq O(\sqrt{T \sum_{u \in Q_{\mathcal{T}}} |\mathsf{ilab}[\mathsf{E}_{\mathcal{T}}[q]]| \log(N)})$.

Moreover, the computational cost of the $t$-th iteration of the algorithm is dominated by $\tau_t$ matrix multiplications or the solution of the linear system. $\tau_t$ can be bounded as follows: $\tau_t = \left\lceil \frac{\log\left(\frac{1}{\sqrt{t}}\right)}{\log(1-\alpha_t)} \right\rceil \leq \frac{\log\left(\frac{1}{\sqrt{t}}\right)}{\log(1-\alpha)} + 1$. Thus, the computational cost of the $t$-th iteration is in

$$O\left( N^2 \min\left\{ \frac{\log t}{\log(1/(1-\alpha_t))}, N \right\} \right) \leq O\left( N^2 \min\left\{ \frac{\log T}{\log(1/(1-\alpha))}, N \right\} \right).$$

$\square$

# F Proof of Theorem 3

**Theorem 3.** *Let $(\mathcal{A}_{I,u,i})_{I\in\mathcal{I}, u\in Q_{\mathcal{T}}, i\in\mathsf{ilab}[\mathsf{E}_{\mathcal{T}}[q]]}$ be external regret minimizing algorithms admitting data-dependent regret bounds of the form $O(\sqrt{L_T(\mathcal{A}_{I,u,i})\log N})$, where $L_T(\mathcal{A}_{I,u,i})$ is the cumulative loss of $\mathcal{A}_{I,u,i}$ after $T$ rounds. Let $\mathcal{A}_{\mathcal{I}}$ be an external regret minimizing algorithm over $\mathcal{I}$ that admits a regret in $O(\sqrt{T\log(|\mathcal{I}|)})$ after $T$ rounds. Assume further that at each round, the sum of the minimal probabilities given to an expert by these algorithms is bounded below by some constant $\alpha > 0$. Then, FASTTIMESELECTTRANSDUCE achieves a time-selection transductive regret with respect to the time-selection family $\mathcal{I}$ and WFST family $\mathcal{T}$ that is in $O\left(\sqrt{T\left(\log(|\mathcal{I}|) + |\mathsf{E}_{\mathcal{T}}|_{\mathsf{in}}\log N\right)}\right)$ with a per-iteration complexity in $O\left(N^2\left(\min\left\{\frac{\log(T)}{\log((1-\alpha)^{-1})}, N\right\} + |\mathcal{I}|\right)\right)$.*

*Proof.* We first note that since $\mathcal{A}_{\mathcal{I}}$ is designed to minimize external regret against the losses $(\tilde{\mathbf{l}}^t)_{t=1}^T$, it follows that for any $I^* \in \mathcal{I}$,

$$\sum_{t=1}^T \sum_{I\in\mathcal{I}} \tilde{\mathsf{q}}_I^t \tilde{l}_I^t \leq \sum_{t=1}^T \tilde{l}_{I^*}^t + \mathrm{Reg}_T(\mathcal{A}_{\mathcal{I}}).$$

Let $u_t$ be the state that the algorithm is in at time $t$ as a result of its past actions. Consider the matrix $\mathbf{Q}^{t,u_t}$ defined in the algorithm. The restriction of the matrix $\mathbf{Q}^{t,u_t}$ to its non-zero rows and columns is a row stochastic matrix. Let $\mathsf{p}_t^*$ be its stationary distribution, and by augmenting it with zeros in the zero rows of $\mathbf{Q}^{t,u_t}$, we may take $\mathsf{p}_t^* \in \Delta_N$ as a fixed point of of $\mathbf{Q}^{t,u_t}$. Then, by expanding the definition of $\tilde{\mathbf{l}}^t$, we can rewrite the expression on the left-hand side as

$$\sum_{t=1}^T \sum_{I\in\mathcal{I}} \tilde{\mathsf{q}}_I^t \tilde{l}_I^t \mathbf{1}_{\tau_t < N} = \sum_{t=1}^T \sum_{I\in\mathcal{I}} \tilde{\mathsf{q}}_I^t I(t) \left(\mathsf{p}_t^\top \mathbf{M}^{t,u_t,I}\mathbf{l}_t - \mathsf{p}_t^\top \mathbf{l}_t\right) \mathbf{1}_{\tau_t < N}$$

$$= \sum_{t=1}^T \sum_{I\in\mathcal{I}} \tilde{\mathsf{q}}_I^t I(t)\mathsf{p}_t^\top \mathbf{M}^{t,u_t,I}\mathbf{l}_t \mathbf{1}_{\tau_t < N} - \sum_{t=1}^T \sum_{I\in\mathcal{I}} \tilde{\mathsf{q}}_I^t I(t)\mathsf{p}_t^\top \mathbf{l}_t \mathbf{1}_{\tau_t < N}$$

$$\geq \sum_{t=1}^T \sum_{I\in\mathcal{I}} \tilde{\mathsf{q}}_I^t I(t)(\mathsf{p}_t^*)^\top \mathbf{M}^{t,u_t,I}\mathbf{l}_t \mathbf{1}_{\tau_t < N} - \sum_{t=1}^T \sum_{I\in\mathcal{I}} \tilde{\mathsf{q}}_I^t I(t)(\mathsf{p}_t^*)^\top \mathbf{l}_t \mathbf{1}_{\tau_t < N}$$

$$- \sum_{t=1}^T \|\mathsf{p}_t - \mathsf{p}_t^*\|_1 \mathbf{1}_{\tau_t < N}.$$

On the other hand, by design, if $\tau_t \geq N$, then $\mathsf{p}_t = \mathsf{p}_t^*$, so that

$$\sum_{t=1}^T \sum_{I\in\mathcal{I}} \tilde{\mathsf{q}}_I^t \tilde{l}_I^t \mathbf{1}_{\tau_t \geq N} = \sum_{t=1}^T \sum_{I\in\mathcal{I}} \tilde{\mathsf{q}}_I^t I(t)(\mathsf{p}_t^*)^\top \mathbf{M}^{t,u_t,I}\mathbf{l}_t \mathbf{1}_{\tau_t \geq N} - \sum_{t=1}^T \sum_{I\in\mathcal{I}} \tilde{\mathsf{q}}_I^t I(t)(\mathsf{p}_t^*)^\top \mathbf{l}_t \mathbf{1}_{\tau_t \geq N}.$$

Thus, it follows that

$$\sum_{t=1}^T \sum_{I\in\mathcal{I}} \tilde{\mathsf{q}}_I^t \tilde{l}_I^t \geq \sum_{t=1}^T \sum_{I\in\mathcal{I}} \tilde{\mathsf{q}}_I^t I(t)(\mathsf{p}_t^*)^\top \mathbf{M}^{t,u_t,I}\mathbf{l}_t - \sum_{t=0}^T \sum_{I\in\mathcal{I}} \tilde{\mathsf{q}}_I^t I(t)(\mathsf{p}_t^*)^\top \mathbf{l}_t - \sum_{t=1}^T \|\mathsf{p}_t - \mathsf{p}_t^*\|_1 \mathbf{1}_{\tau_t < N}.$$

If $\sum_{I\in\mathcal{I}} I(t)\tilde{\mathsf{q}}_I^t \neq 0$, then the fact that $\mathsf{p}_t^*$ is a stationary distribution of $\mathbf{Q}^t = \frac{\sum_{I\in\mathcal{I}} I(t)\tilde{\mathsf{q}}_I^t \mathbf{M}^{t,u_t,I}}{\sum_{I\in\mathcal{I}} I(t)\tilde{\mathsf{q}}_I^t}$ implies that

$$\sum_{I\in\mathcal{I}} \tilde{\mathsf{q}}_I^t I(t)(\mathsf{p}_t^*)^\top \mathbf{M}^{t,u_t,I}\mathbf{l}_t = \sum_{I\in\mathcal{I}} \tilde{\mathsf{q}}_I^t I(t)(\mathsf{p}_t^*)^\top \mathbf{l}_t.$$

On the other hand, if $\sum_{I\in\mathcal{I}} I(t)\tilde{\mathsf{q}}_I^t = 0$, then by non-negativity, it must be the case that $I(t)\tilde{\mathsf{q}}_I^t = 0$ for every $I \in \mathcal{I}$. Thus, it follows that

$$\sum_{I\in\mathcal{I}} \tilde{\mathsf{q}}_I^t I(t)(\mathsf{p}_t^*)^\top \mathbf{M}^{t,u_t,I}\mathbf{l}_t = \sum_{I\in\mathcal{I}} \tilde{\mathsf{q}}_I^t I(t)(\mathsf{p}_t^*)^\top \mathbf{l}_t = 0,$$

which implies that

$$\sum_{t=1}^{T} -\tilde{l}_{I^*}^t \leq \sum_{t=1}^{T} \|\mathsf{p}_t - \mathsf{p}_t^*\|_1 1_{\tau_t < N} + \mathrm{Reg}_T(\mathcal{A}_\mathcal{I}).$$

By expanding the definition of $\tilde{l}_{I^*}^t$, we can write

$$\sum_{t=1}^{T} -\tilde{l}_{I^*}^t = \sum_{t=1}^{T} -I^*(t)\left(\mathsf{p}_t^\top \mathbf{M}^{t,u_t,I^*}\mathbf{l}_t - \mathsf{p}_t^\top \mathbf{l}_t\right) = \sum_{t=1}^{T} I^*(t)\mathsf{p}_t^\top \mathbf{l}_t - I^*(t)\mathsf{p}_t^\top \mathbf{M}^{t,u_t,I^*}\mathbf{l}_t.$$

Moreover, for any $\mathcal{T} \in \mathcal{T}$, we can bound the second term in the following way:

$$\sum_{t=1}^{T} I^*(t)\mathsf{p}_t^\top \mathbf{M}^{t,u_t,I^*}\mathbf{l}_t = \sum_{t=1}^{T} I^*(t) \sum_{i=1}^{N} \mathsf{p}_{t,i} \sum_{j=1}^{N} \mathbf{M}_{i,j}^{t,u_t,I^*} l_{t,j}$$

$$= \sum_{u \in Q_\mathcal{T}} \sum_{i=1}^{N} \sum_{t=1}^{T} \sum_{j=1}^{N} \mathbf{M}_{i,j}^{t,u_t,I^*} 1_{\delta_\mathcal{T}(I_\mathcal{T},x_{1:t-1})=u} I^*(t)\mathsf{p}_{t,i} l_{t,j}$$

$$= \sum_{u \in Q_\mathcal{T}} \sum_{i \in \mathsf{ilab}[\mathsf{E}_\mathcal{T}[u]]} \sum_{t=1}^{T} \sum_{j=1}^{N} \mathbf{M}_{i,j}^{t,u_t,I^*} 1_{\delta_\mathcal{T}(I_\mathcal{T},x_{1:t-1})=u} I^*(t)\mathsf{p}_{t,i} l_{t,j}$$

$$\leq \sum_{u \in Q_\mathcal{T}} \sum_{i \in \mathsf{ilab}[\mathsf{E}_\mathcal{T}[u]]} \min_{i^* \in \mathsf{olab}[\mathsf{E}_\mathcal{T}[u]]} \sum_{t=1}^{T} 1_{\delta_\mathcal{T}(I_\mathcal{T},x_{1:t-1})=u} I^*(t)\mathsf{p}_{t,i} l_{t,i^*}$$

$$+ \sum_{u \in Q_\mathcal{T}} \sum_{i \in \mathsf{ilab}[\mathsf{E}_\mathcal{T}[u]]} \mathrm{Reg}_T(\mathcal{A}_{I,u,i}, \Phi_{\mathrm{ext}})$$

$$\leq \sum_{u \in Q_\mathcal{T}} \sum_{i \in \mathsf{ilab}[\mathsf{E}_\mathcal{T}[u]]} \sum_{e \in \mathsf{E}_\mathcal{J}[u]} w[e] \sum_{t=1}^{T} 1_{\delta_\mathcal{T}(I_\mathcal{T},x_{1:t-1})=u} I^*(t)\mathsf{p}_{t,i} l_{t,\mathsf{olab}[e]}$$

$$+ \sum_{u \in Q_\mathcal{T}} \sum_{i \in \mathsf{ilab}[\mathsf{E}_\mathcal{J}[u]]} \mathrm{Reg}_T(\mathcal{A}_{I,u,i}, \Phi_{\mathrm{ext}})$$

$$= \sum_{t=1}^{T} I^*(t) \mathop{\mathbb{E}}_{x_t \sim \mathsf{p}_t} \left[ \sum_{e \in \mathsf{E}_\mathcal{J}[\delta_\mathcal{T}(I_\mathcal{T},x_{1:t-1}),x_t]} w[e] l_t(\mathsf{olab}[e]) \right]$$

$$+ \sum_{u \in Q_\mathcal{T}} \sum_{i \in \mathsf{ilab}[\mathsf{E}_\mathcal{T}[u]]} \mathrm{Reg}_T(\mathcal{A}_{I,u,i}, \Phi_{\mathrm{ext}}),$$

using the fact that algorithm $\mathcal{A}_{I,u,i}$ minimizes external regret against the surrogate losses $I(t)1_{\delta_\mathcal{T}(I_\Phi,x_{1:t-1})=u}\mathsf{p}_{t,i}\mathbf{l}_t$.

As in Theorem 2, the scaling assumption on the external regret minimizing algorithms and Jensen's inequality imply that

$$\sum_{u \in Q_\mathcal{T}} \sum_{i \in \mathsf{ilab}[\mathsf{E}_\mathcal{T}[u]]} \mathrm{Reg}_T(\mathcal{A}_{I,u,i}, \Phi_{\mathrm{ext}}) \leq O\left( \sqrt{T \sum_{u \in Q_\mathcal{T}} |\mathsf{ilab}[\mathsf{E}_\mathcal{J}[u]]| \log(N)} \right).$$

Thus, we can write for any $I^* \in \mathcal{I}$ that

$$\sum_{t=1}^{T} I^*(t)\mathsf{p}_t^\top \mathbf{l}_t - I^*(t)\mathsf{p}_t^\top \mathbf{M}^{t,u_t,I^*}\mathbf{l}_t - \sum_{t=1}^{T} I^*(t) \mathop{\mathbb{E}}_{x_t \sim \mathsf{p}_t} \left[ \sum_{e \in \mathsf{E}_\mathcal{J}[\delta_\mathcal{T}(I_\mathcal{T},x_{1:t-1}),x_t]} w[e] l_t(\mathsf{olab}[e]) \right]$$

$$\leq \mathrm{Reg}_T(\mathcal{A}_\mathcal{I}) + O\left( \sqrt{T \sum_{u \in Q_\mathcal{T}} |\mathsf{ilab}[\mathsf{E}_\mathcal{J}[u]]| \log(N)} \right) + \sum_{t=1}^{T} \|\mathsf{p}_t - \mathsf{p}_t^*\|_1 1_{\tau_t < N},$$

and as in Theorem 2, we can bound the $l_1$ approximation error of $\mathsf{p}_t$ for $\mathsf{p}_t^*$ by

$$\|\mathsf{p}_t - \mathsf{p}_t^*\|_1 \leq (1 - \alpha_t)^{\tau_t} \leq \frac{1}{\sqrt{t}},$$

by the algorithm's choice of $\tau_t$. Thus, by applying regret guarantee of algorithm $\mathcal{A}_\mathcal{I}$ together with the above calculations, the time-selection transductive regret of FASTTIMESELECTTRANSDUCE is in $O\left(\sqrt{T\left(\log(|\mathcal{I}|) + \sum_{q \in Q_\Phi} |\mathsf{ilab}[\mathsf{E}_\mathcal{T}[q]]| \log(N)\right)}\right)$.

Moreover, at each round $t$, the computational cost of the algorithm is dominated by two quantities: the update of $|\mathcal{I}|N$ external regret minimizing algorithms over the $N$ experts, which is in $O(|\mathcal{I}|N^2)$, and the fixed-point approximation or solution of the linear system, which is in

$$O\left(N^2 \min\left\{\frac{\log(t)}{\log\left((1-\alpha_t)^{-1}\right)}, N\right\}\right) \leq O\left(N^2 \min\left\{\frac{\log(T)}{\log\left((1-\alpha)^{-1}\right)}, N\right\}\right).$$

$\square$

# G  Proof of Theorem 4

**Theorem 4.** *Assume that the sleeping regret minimizing algorithms used as inputs of* FASTSLEEPTRANSDUCE *achieve data-dependent regret bounds such that, if the algorithm selects the distributions $(\mathsf{p}_t)_{t=1}^T$ and observes losses $(\mathbf{l}_t)_{t=1}^T$ with awake sets $(A_t)_{t=1}^T$, then the regret of $\mathcal{A}_i^q$ is at most $O\left(\sqrt{\sum_{t=1}^T \mathsf{u}^*(A_t)\,\mathbb{E}_{x_t \sim \mathsf{p}_t}[l_t(x_t)]\log(N)}\right)$. Assume further that at each round, the sum of the minimal probabilities given to an expert by these algorithms is bounded below by some constant $\alpha > 0$. Then, the sleeping regret $\mathrm{Reg}_T(\text{FASTSLEEPTRANSDUCE}, \mathcal{T}, A_1^T)$ of* FASTSLEEPTRANS- *DUCE is upper bounded by $O\left(\sqrt{\sum_{t=1}^T \mathsf{u}(A_t)|\mathsf{E}_{\mathcal{T}}|_{\mathsf{in}}\log(N)}\right)$. Moreover,* FASTSLEEPTRANSDUCE *admits a per-iteration complexity in $O\left(N^2 \min\left\{\frac{\log T}{\log(1/(1-\alpha))}, N\right\}\right)$.*

*Proof.* Let $\mathsf{u} \in \Delta_N$, and let $\mathsf{p}_t^{A_t}$ be the distribution output by FASTSLEEPTRANSDUCE at round $t$. For any distribution $\mathsf{p}_t^*$, $t \in [T]$, the following inequalities hold:

$$
\mathsf{u}(A_t) \underset{x_t \sim \mathsf{p}_t^{A_t}}{\mathbb{E}}[l_t(x_t)]1_{\tau_t < N} = \mathsf{u}(A_t)\left(\underset{x_t \sim \mathsf{p}_t^{A_t,*}}{\mathbb{E}}[l_t(x_t)] + \underset{x_t \sim \mathsf{p}_t^{A_t}}{\mathbb{E}}[l_t(x_t)] - \underset{x_t \sim \mathsf{p}_t^{A_t,*}}{\mathbb{E}}[l_t(x_t)]\right)1_{\tau_t < N}
$$

$$
\leq \mathsf{u}(A_t)\left(\underset{x_t \sim \mathsf{p}_t^{A_t,*}}{\mathbb{E}}[l_t(x_t)] + \|\mathsf{p}_t^{A_t} - \mathsf{p}_t^{A_t,*}\|_1 \|l_t\|_\infty\right)1_{\tau_t < N}
$$

$$
\leq \mathsf{u}(A_t)\left(\underset{x_t \sim \mathsf{p}_t^{A_t,*}}{\mathbb{E}}[l_t(x_t)] + \|\mathsf{p}_t^{A_t} - \mathsf{p}_t^{A_t,*}\|_1\right)1_{\tau_t < N}.
$$

Let $u_t$ be the state that the algorithm is in at time $t$ as a result of its past actions. Consider the matrix $\mathbf{Q}^{t,u_t}$ defined in the algorithm. The restriction of $\mathbf{Q}^{t,u_t}$ to its non-zero rows and columns is a row stochastic matrix. Let $\mathsf{p}_t^{A_t,*}$ be its stationary distribution, and by augmenting it with zeros in the zero rows of $\mathbf{Q}^{t,u_t}$, we may take $\mathsf{p}_t^{A_t,*} \in \Delta_N$ as a fixed point of $\mathbf{Q}^{t,u_t}$. Then, we can write:

$$
\sum_{t=1}^T \mathsf{u}(A_t) \underset{x_t \sim \mathsf{p}_t^{A_t,*}}{\mathbb{E}}[l_t(x_t)]1_{\tau_t < N}
$$

$$
= \sum_{t=1}^T \sum_{i=1}^N \sum_{j=1}^N \mathsf{u}(A_t)\mathsf{p}_{t,i}^{A_t,*}\mathbf{Q}_{i,j}^{t,u_t}l_{t,j}1_{\tau_t < N}
$$

$$
= \sum_{i=1}^N \sum_{t=1}^T \sum_{j=1}^N \mathsf{u}(A_t)\mathbf{Q}_{i,j}^{t,u_t}\mathsf{p}_{t,i}^{A_t}l_{t,j}1_{\tau_t < N} + \sum_{i=1}^N \sum_{t=1}^T \sum_{j=1}^N \mathsf{u}(A_t)\mathbf{Q}_{i,j}^{t,u_t}(\mathsf{p}_{t,i}^{A_t,*} - \mathsf{p}_{t,i}^{A_t})l_{t,j}1_{\tau_t < N}
$$

$$
\leq \sum_{i=1}^N \sum_{t=1}^T \sum_{j=1}^N \mathsf{u}(A_t)\mathbf{Q}_{i,j}^{t,u_t}\mathsf{p}_{t,i}^{A_t}l_{t,j}1_{\tau_t < N} + \sum_{t=1}^T \mathsf{u}(A_t)\|\mathsf{p}_t^{A_t,*} - \mathsf{p}_t^{A_t}\|_1 1_{\tau_t < N}.
$$

On the other hand, by design, if $\tau_t \geq N$, then $\mathsf{p}_t = \mathsf{p}_t^*$, so that

$$
\sum_{t=1}^T \mathsf{u}(A_t) \underset{x_t \sim \mathsf{p}_t^{A_t}}{\mathbb{E}}[l_t(x_t)]1_{\tau_t \geq N} \leq \sum_{t=1}^T \sum_{i=1}^N \sum_{t=1}^T \sum_{j=1}^N \mathsf{u}(A_t)\mathbf{Q}_{i,j}^{t,u_t}\mathsf{p}_{t,i}^{A_t}l_{t,j}1_{\tau_t \geq N}.
$$

Thus, it follows that for any WFST $\mathcal{T} \in \mathcal{T}$,

$$
\sum_{t=1}^T \mathsf{u}(A_t) \underset{x_t \sim \mathsf{p}_t^{A_t}}{\mathbb{E}}[l_t(x_t)]
$$

$$
\leq \sum_{i=1}^N \sum_{t=1}^T \sum_{j=1}^N \mathsf{u}(A_t)\mathbf{Q}_{i,j}^{t,u_t}\mathsf{p}_{t,i}^{A_t}l_{t,j} + 2\sum_{t=1}^T \mathsf{u}(A_t)\|\mathsf{p}_t^{A_t,*} - \mathsf{p}_t^{A_t}\|_1 1_{\tau_t < N}
$$

$$= \sum_{i=1}^{N}\sum_{t=1}^{T}\sum_{j=1}^{N}\sum_{u\in Q_{\mathcal{T}}} \mathsf{u}(A_t)\mathbf{Q}_{i,j}^{t,u}1_{\delta_{\mathcal{T}}(I_{\mathcal{T}},x_{1:t-1})=u}\mathsf{p}_{t,i}^{A_t}l_{t,j} + 2\sum_{t=1}^{T}\mathsf{u}(A_t)\|\mathsf{p}_t^{A_t,*}-\mathsf{p}_t^{A_t}\|_1 1_{\tau_t<N}$$

$$= \sum_{u\in Q_{\mathcal{T}}}\sum_{i\in\mathsf{ilab}[\mathsf{E}_{\mathcal{T}}[u]]}\sum_{t=1}^{T}\sum_{j=1}^{N}\mathsf{u}(A_t)\mathbf{Q}_{i,j}^{t,u}1_{\delta_{\mathcal{T}}(I_{\mathcal{T}},x_{1:t-1})=u}\mathsf{p}_{t,i}^{A_t}l_{t,j}$$

$$+ 2\sum_{t=1}^{T}\mathsf{u}(A_t)\|\mathsf{p}_t^{A_t,*}-\mathsf{p}_t^{A_t}\|_1 1_{\tau_t<N}$$

$$\le \sum_{u\in Q_{\mathcal{T}}}\sum_{i\in\mathsf{ilab}[\mathsf{E}_{\mathcal{T}}[u]]}\min_{\substack{\mathsf{u}^{u,i}\in\Delta_N\\ \sum_{j\in A_t}\mathsf{u}_j^{q,i}=\mathsf{u}(A_t)}}\sum_{t=1}^{T}\sum_{j=1}^{N}1_{\delta_{\mathcal{T}}(I_{\mathcal{T}},x_{1:t-1})=u}\mathsf{u}_j^{q,i}1_{j\in A_t}\mathsf{p}_{t,i}^{A_t}l_{t,j}$$

$$+ 2\sum_{t=1}^{T}\mathsf{u}(A_t)\|\mathsf{p}_t^{A_t,*}-\mathsf{p}_t^{A_t}\|_1 1_{\tau_t<N} + \sum_{u\in Q_{\mathcal{T}}}\sum_{i\in\mathsf{ilab}[\mathsf{E}_{\mathcal{T}}[u]]}\mathrm{Reg}_T(\mathcal{A}_{u,i},\Phi_{\mathrm{sleep}})$$

$$\le \sum_{u\in Q_{\mathcal{T}}}\sum_{i\in\mathsf{ilab}[\mathsf{E}_{\mathcal{T}}[u]]}\sum_{e\in\mathsf{E}_{\mathcal{T}}[q]}\sum_{t=1}^{T}\sum_{j=1}^{N}1_{\delta_{\mathcal{T}}(I_{\mathcal{T}},x_{1:t-1})=u}\mathsf{u}_j 1_{j\in A_t}w[e]\mathsf{p}_{t,i}^{A_t}l_{t,j}$$

$$+ 2\sum_{t=1}^{T}\mathsf{u}(A_t)\|\mathsf{p}_t^{A_t,*}-\mathsf{p}_t^{A_t}\|_1 1_{\tau_t<N} + \sum_{u\in Q_{\mathcal{T}}}\sum_{i\in\mathsf{ilab}[\mathsf{E}_{\mathcal{T}}[q]]}\mathrm{Reg}_T(\mathcal{A}_{u,i},\Phi_{\mathrm{sleep}})$$

$$= \sum_{t=1}^{T}\mathbb{E}_{x_t\sim\mathsf{p}_t}\left[\sum_{e\in\mathsf{E}_{\mathcal{T}}[\delta_{\mathcal{T}}(I_{\mathcal{T}},x_{1:t-1}),x_t]}(\mathsf{u}|_{A_t})_{\mathsf{olab}[e]}w[e]\mathsf{p}_{t,i}^{A_t}l_t(\mathsf{olab}[e])\right]$$

$$+ 2\sum_{t=1}^{T}\mathsf{u}(A_t)\|\mathsf{p}_t^{A_t,*}-\mathsf{p}_t^{A_t}\|_1 1_{\tau_t<N} + \sum_{u\in Q_{\mathcal{T}}}\sum_{i\in\mathsf{ilab}[\mathsf{E}_{\mathcal{T}}[q]]}\mathrm{Reg}_T(\mathcal{A}_{u,i},\Phi_{\mathrm{sleep}}).$$

For any distribution $\mathsf{u}^*\in\Delta_N$ and awake sequence $A_1^T$, Let $L_T^{\mathsf{u},A_1^T}=\sum_{t=1}^{T}\mathsf{u}^*(A_t)\mathbb{E}_{x_t\sim\mathsf{p}_t}[l_t(x_t)]$, Thus, algorithm $\mathcal{A}_{u,i}$ achieves a regret in $O(\sqrt{L_T^{\mathsf{u}_i^{q,*},A_1^T}\log(N)})$, where $\mathsf{u}_i^{q,*}$ is a maximizer of algorithm $\mathcal{A}_{u,i}$'s sleeping regret.

Since the losses attributed to algorithm $\mathcal{A}_{u,i}$ are scaled by $1_{\delta_{\mathcal{T}}(I_{\mathcal{T}},x_{1:t-1})=u}\mathsf{p}_{t,i}^{A_t}$, it follows that at each round, the sum of the losses over all the algorithms is at most 1. Thus, by Jensen's inequality, it follows that

$$\frac{1}{\sum_{u\in Q_{\mathcal{T}}}|\mathsf{ilab}[\mathsf{E}_{\mathcal{T}}[u]]|}\sum_{u\in Q_{\mathcal{T}}}\sum_{i\in\mathsf{ilab}[\mathsf{E}_{\mathcal{T}}[u]]}\mathrm{Reg}_T(\mathcal{A}_{u,i},\Phi_{\mathrm{sleep}})$$

$$= \frac{1}{\sum_{u\in Q_{\mathcal{T}}}|\mathsf{ilab}[\mathsf{E}_{\mathcal{T}}[u]]|}\sum_{u\in Q_{\mathcal{T}}}\sum_{i\in\mathsf{ilab}[\mathsf{E}_{\mathcal{T}}[u]]}\sqrt{L_T^{\mathsf{u}_i^{q,*},A_1^T}(\mathcal{A}_{u,i})\log(N)}$$

$$\le \sqrt{\frac{1}{\sum_{u\in Q_{\mathcal{T}}}|\mathsf{ilab}[\mathsf{E}_{\mathcal{T}}[u]]|}\sum_{u\in Q_{\mathcal{T}}}\sum_{i\in\mathsf{ilab}[\mathsf{E}_{\mathcal{T}}[u]]}L_T^{\mathsf{u},A_1^T}(\mathcal{A}_{u,i})\log(N)}$$

$$\le \sqrt{\frac{1}{\sum_{u\in Q_{\mathcal{T}}}|\mathsf{ilab}[\mathsf{E}_{\mathcal{T}}[u]]|}\sum_{t=1}^{T}\mathsf{u}(A_t)\log(N)},$$

so that $\sum_{u\in Q_{\mathcal{T}}}\sum_{i\in\mathsf{ilab}[\mathsf{E}_{\mathcal{T}}[u]]}\mathrm{Reg}_T(\mathcal{A}_{u,i},\Phi_{\mathrm{sleep}})\le\sqrt{\sum_{t=1}^{T}\mathsf{u}(A_t)\sum_{u\in Q_{\mathcal{T}}}\sum_{i\in\mathsf{ilab}[\mathsf{E}_{\mathcal{T}}[u]]}\log(N)}$.

Finally, during the rounds in which $1_{\tau_t<N}$, $\mathsf{p}_t$ is an RPM approximation of $\mathsf{p}_t^*$ using $\tau_t$ iterations. Thus, by Equation 3.7 in [Nesterov and Nemirovski, 2015] the following inequality holds: $\|\mathsf{p}_t - \mathsf{p}_t^*\|_1 \le (1-\alpha_t)^{\tau_t}$. Since $\tau_t$ is chosen so that the inequality $(1-\alpha_t)^{\tau_t}\le 1/\sqrt{t}$ holds, it follows that

$\sum_{t=1}^{T} \mathsf{u}(A_t) \| \mathsf{p}_t^{A_t} - \mathsf{p}_t^{A_t,*} \|_1 \leq \sqrt{T}$, which proves the regret bound

$$\sum_{t=1}^{T} \mathsf{u}(A_t) \mathop{\mathbb{E}}_{x_t \sim \mathsf{p}_t^{A_t}} [l_t(x_t)] - \sum_{t=1}^{T} \mathop{\mathbb{E}}_{x_t \sim \mathsf{p}_t^{A_t}} \left[ \sum_{e \in \mathsf{E}_{\mathcal{T}}[\delta_{\mathcal{T}}(I_{\mathcal{T}}, x_{1:t-1}), x_t]} (\mathsf{u}|_{A_t})_{\mathsf{olab}[e]} w[e] l_t(\mathsf{olab}[e]) \right]$$

$$\leq O \left( \sqrt{\sum_{t=1}^{T} \mathsf{u}(A_t) \sum_{q \in Q_\Phi} \sum_{i \in \mathsf{ilab}[\mathsf{E}_{\mathcal{T}}[q]]} \log(N)} \right).$$

Furthermore, the computational cost of the $t$-th iteration of the algorithm is dominated by $\tau_t$ matrix multiplications or the solution of the linear system. $\tau_t$ can be bounded as follows: $\tau_t = \left\lceil \frac{\log\left(\frac{1}{\sqrt{t}}\right)}{\log(1-\alpha_t)} \right\rceil \leq \frac{\log\left(\frac{1}{\sqrt{t}}\right)}{\log(1-\alpha)} + 1$. Thus, the computational cost of the $t$-th iteration is in

$$O \left( N^2 \min\left\{ \frac{\log t}{\log(1/(1-\alpha_t))}, N \right\} \right) \leq O \left( N^2 \min\left\{ \frac{\log T}{\log(1/(1-\alpha))}, N \right\} \right).$$

□

## H Connections with game-theoretic equilibria

There is an elegant connection between regret minimization in online learning and convergence to game-theoretic equilibria in repeated games [Nisan et al., 2007]. As an example, remarkably, if all players in a repeated game follow a swap regret minimization algorithm, then the empirical distribution of their play converges to a correlated equilibrium (see for example [Blum and Mansour, 2007]). Similarly, if all players follow a conditional swap regret minimization algorithm, then the empirical distribution of their play converges to a conditional correlated equilibrium [Mohri and Yang, 2014]. Hazan and Kale [2008] showed a result generalizing this property to the case of a $\Phi$-regret and $\Phi$-equilibrium. Moreover, the authors showed that the existence of an efficient $\Phi$-regret minimizing algorithm is equivalent to the possibility of efficiently computing a fixed point associated to $\Phi$-regret. However, their characterization of efficiency is a per iteration time complexity of $O(|\Phi|)$, which may be very large, in fact exponential in the number of experts, as in the case of the examples discussed in this paper. Here, we proved the existence of a large class of $\Phi$-equilibria, *transductive equilibria*, i.e. those induced by a WFST, that are realizable in time that is polynomial in the number of experts.

## I Lower bound

Auer [2017] proved a lower bound of $\Omega(\sqrt{TN})$ for swap regret. Since swap regret is a special case of transductive regret, that lower bound applies to the setting of transductive regret as well. This is further detailed in an extended version of this paper.

## J Bandit setting

Blum and Mansour [2007] and Mohri and Yang [2014] respectively showed that swap and conditional swap regret-minimizing algorithms can be extended to the bandit setting. Similarly, our more general transductive regret-minimizing can be extended to the bandit setting, as shown and detailed in the extended version of this paper.