[Reviews · NeurIPS 2017]

Reviewer 1



The paper considers the classic setting of prediction with expert advice and proposes a new notion of regret called "transductive regret", which measures the performance of the learner against the best from a class of finite-state transducers. Precisely, a transducer will read the history of past predictions of the learner and output a distribution over the next predictions; for forming the next prediction, the learner can take advantage of the class of transducers under consideration. The resulting notion of regret generalizes a number of known settings such as external and internal regret, swap regret and conditional swap regret. The main contribution is proposing an algorithm that achieves a regret of at most sqrt(E*T), where E is bounded by the number of transitions in the considered class of transducers. The authors also extend their algorithm to other settings involving time-selection functions and sleeping experts. The paper is very technical and seems to target a rather specialized audience. The considered setting is very general and highly abstract---which is actually my main concern about the paper. Indeed, the authors fail to motivate their very general setting appropriately: the only justification given is that the new notion generalizes the most general related regret notion so far, that of conditional swap regret. There is one specific example in Section 4.2 that was not covered by previous results, but I'm not sure I understand why this particular regret notion would be useful. Specifically, I fail to see how exactly is the learner "penalized" by having to measure its regret against a restricted set of experts after having picked another expert too many times---if anything, measuring regret against a smaller class just makes the problem *easier* for the learner. Indeed, it is trivial to achieve bounded regret in this setting: just choose "b" n times and "a" in all further rounds, thus obtaining a regret of exactly n, independently of the number of rounds T. The paper would really benefit from providing a more convincing motivating example. The paper is easy to read for the most part, with the critical exception of Section 4.1 that describes the concept of weighted finite-state transducers (WFSTs). This part was quite confusing for me to read, specifically because it is never explicitly revealed that in the specific online learning setting, labels will correspond to experts and the states will remain merely internal variables of WFSTs. Actually, the mechanism of how a WFST operates and produces output labels is never described either---while the reader can certainly figure this out after a bit of thought, I think a few explanatory sentences would be useful to remove ambiguity. The fact that the letters E and T are so heavily overloaded does not help readability either. The analysis techniques appear to be rather standard, with most technical tools apparently borrowed from Blum and Mansour (2007). However, this is difficult to tell as the authors do not provide any discussion in the main text about the intuition underlying the analysis or the crucial difficulties that needed to be overcome to complete the proofs. The authors also provide some computational improvements over the standard reduction of Blum and Mansour (2007) and its previous improved variants, but these improvements are, in my view, minor. Actually, at the end of Section 5, the authors claim an exponential improvement over a naive implementation, but this seems to contradict the previous discussion at the beginning of the section about the work of Khot and Ponnuswami (2008). I would also like to call the work of Maillard and Munos (ICML 2011) into the authors' attention: they also study a similar notion of regret relative to an even more general class of functions operating over histories of actions taken by the learner. This work is definitely relevant to the current paper and the similarities/differences beg to be discussed. Overall, while I appreciate the generality of the considered setting and the amount of technical work invested into proving the main results, I am not yet convinced about their usefulness. Indeed, I can't really see the potential impact of this work in lack of one single example where the newly proposed regret notion is more useful than others. I hope to be convinced by the author response so that I can vote for acceptance---right now, I'm leaning a little bit towards suggesting rejection. Detailed comments ================= 052: Notation is somewhat inconsistent, e.g., the loss function l_t is sometimes in plain typeface, sometimes bold... 055: Missing parens in the displayed equation (this is a recurring error throughout the paper). 093: Is it really important to have O(\sqrt{L_T}) regret bounds for the base learner? After inspecting the proofs, requesting O(\sqrt{T}) seems to be enough. In fact, this would allow setting alpha = 1/\sqrt{T} artificially for every base learner and would make the statement of all theorems a bit simpler. 099: "t is both the number of rounds and the number of iterations"---I don't understand; I thought the number of iterations was \tau_t = poly(t)? Comments after rebuttal ======================= Thanks for the response. I now fully appreciate the improvement in the results of Section 5, and understand the algorithmic contribution a little better. I still wish that the paper would do a better job in describing the algorithmic and proof techniques, and also in motivating the study a little better. I'm still not entirely sold on the new motivating example provided in the response, specifically: how does such a benchmark "penalize" a learning algorithm and how is this example different from the one by Mohri and Yang (2014) used to motivate conditional swap regret? I raise my score as a sign of my appreciation, but note again that the authors have to work quite a bit more to improve the presentation in the final version.

Reviewer 2



There are multiple interesting contributions in this paper. First, the algorithm FastSwap matches a swap regret bound with better per-iteration complexity. Second, the transductive regret setting is introduced along with multiple algorithms for the setting, composing with time-selection and sleeping regret settings. The exposition is extremely intelligible, the execution thorough, and the results interesting. I suspect the inequality on the first line after line 352 is incorrect, but then the inequality on the second line after line 352 looks correct. Also, why switch between L1-norm notation (block after line 348) and an explicit sum over i (block after line 350)? I feel like Harrison Bergeron. Thankfully the blocks after line 371 and line 375 both use L1-norm notation, but now there is an extra subscripting-i in the second term (?) which is apparently copy-pasted along. Typos aside, I think the proof technique is sound: hitting with the stochastic row is at most the maximum, which is bounded by the best fixed choice plus the slave algorithm regret, and the latter is amenable to Jensen's inequality; furthermore the L1 distance is additive to the regret and can be bounded via RPM without incurring the full O(N^3).

Reviewer 3



The paper presents a new notion of regret called transductive regret that generalizes existing regrets. The key is that the authors allow an arbitrary set of weighted finite-state transducers, which allows flexible design of 'swaps' and generalizes external, internal, swap, and conditional swap regrets. The authors first present an efficient algorithm for swap regret, which makes use of a recent development in power iteration methods called RPM. The computational benefit comes with a mild assumption, but the authors successfully defend it by resorting their algorithm back to the standard one, which allows being 'never slower than prior work'. Next, this core technique is repeatedly applied to enjoy low regret in transductive regret, time-selection transductive regret, and sleeping transductive regret. Overall, the paper is easy to read. The paper sets up a powerful generalization of existing regrets and present algorithms that are efficient, which I believe provides enough novelty. On the other hand, the computational benefit is only 'fast when it can' and I am not sure how much benefit one gets in practice. I would love to see at least in a toy experiments what kind of alpha values one can expect. Details: * I hope to see some motivations for moving beyond the popular external regret. I know this topic has been studied for a while, but still not intuitive and needs some work to broaden the audience. At least, the authors could refer to a few papers that discuss the benefits and applications of these swap-based regrets. * L126: what is \Sigma^*? * L120: the acronym WFST is not defined. could just put the definition and save reader's time. * L256: "arbitrary losses" -> in what sense? beyond linear loss function?